# DynAlign: Unsupervised Dynamic Taxonomy Alignment for Cross-Domain Segmentation

**Han Sun**[1]  **Rui Gong**[2]*  **Ismail Nejjar**[1]  **Olga Fink**[1]
[1]EPFL  [2]Amazon
{han.sun, ismail.nejjar, olga.fink}@epfl.ch   ruigong@amazon.com

## Abstract

Current unsupervised domain adaptation (UDA) methods for semantic segmentation typically assume identical class labels between the source and target domains. This assumption ignores the label-level domain gap, which is common in real-world scenarios, thus limiting their ability to identify finer-grained or novel categories without requiring extensive manual annotation. A promising direction to address this limitation lies in recent advancements in foundation models, which exhibit strong generalization abilities due to their rich prior knowledge. However, these models often struggle with domain-specific nuances and underrepresented fine-grained categories. To address these challenges, we introduce DynAlign , a framework that integrates UDA with foundation models to bridge both the image-level and label-level domain gaps. Our approach leverages prior semantic knowledge to align source categories with target categories that can be novel, more fine-grained, or named differently (e.g., *'vehicle'* to {*'car'*, *'truck'*, *'bus'*}). Foundation models are then employed for precise segmentation and category reassignment. To further enhance accuracy, we propose a knowledge fusion approach that dynamically adapts to varying scene contexts. DynAlign generates accurate predictions in a new target label space without requiring any manual annotations, allowing seamless adaptation to new taxonomies through either model retraining or direct inference. Experiments on the street scene semantic segmentation benchmarks GTA→Mapillary Vistas and GTA→IDD validate the effectiveness of our approach, achieving a significant improvement over existing methods. Our code is publically available at https://github.com/hansunhayden/DynAlign.

## 1 Introduction

Semantic segmentation is a crucial computer vision task that assigns category labels to each pixel in an image, enabling detailed scene understanding. Driven by advancements in deep learning, the field has recently seen significant progress, with applications ranging from autonomous driving (Cheng et al., 2022; Jain et al., 2023) to medical image diagnosis (Cao et al., 2022). Despite this progress, models trained on labeled source datasets often struggle to generalize to data with different distributions due to variations in weather, illumination, or object appearance, resulting in degraded performance. While re-training or fine-tuning models can address this issue, these approaches require annotated in-domain data, which is particularly costly for semantic segmentation.

Unsupervised Domain Adaptation (UDA) addresses the challenge of adapting a model trained on a labeled source domain to an unlabeled target domain by mitigating the adverse effects of domain shift without requiring costly data annotations. However, most UDA are constrained by domain-specific knowledge from the available datasets and typically operate under closed-set assumptions, where the label spaces of the two domains are identical. This assumption limits their applicability in real-world scenarios, where source and target datasets often exhibit the label-level taxonomy gap - including variations in class categories, semantic contexts, and category granularity.

To overcome this limitation, open-set Domain Adaptation (DA) methods have been developed to recognize novel classes in the target domain (Saito & Saenko, 2021; Li et al., 2023). However, these methods are only capable of distinguishing *unknown* classes from *known* ones and do not provide detailed classifications for new target classes. To solve this, taxonomy-adaptive domain

---

*This work was done before Rui Gong joined Amazon.

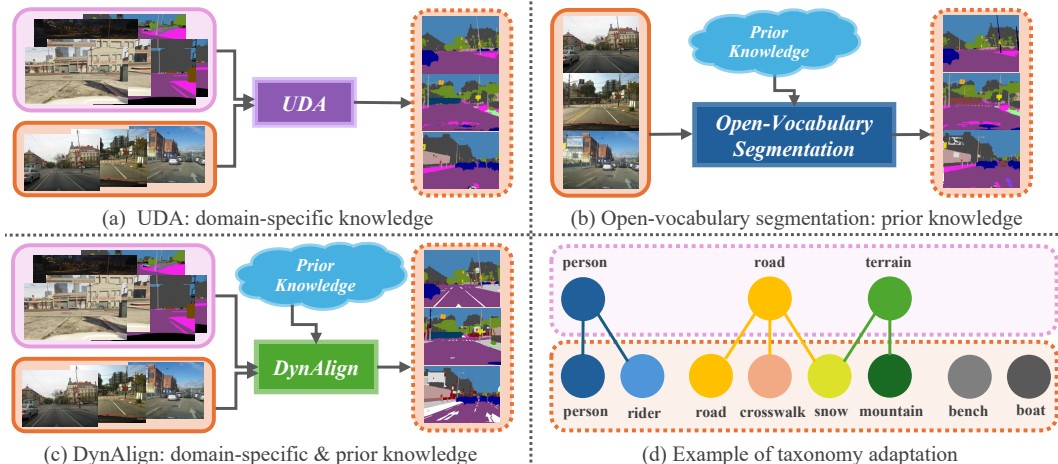

Figure 1: **DynAlign and taxonomy adaptation.** Current UDA methods focus solely on domain-specific knowledge transfer and assume consistent class labels across domains, limiting their flexibility in adapting to different taxonomies. Open-vocabulary segmentation models excel with broader taxonomies through large-scale pretraining but lack the precision of domain-specific models for specialized tasks. In contrast, DynAlign integrates with any UDA model and flexibly adapts to diverse taxonomies and scene contexts, leveraging the prior knowledge of foundational models.

adaptation has been proposed, enabling UDA in settings where the target domain adopts a label space different from that of the source domain (Gong et al., 2022; Fan et al., 2023a). These methods aim to align differing taxonomies by leveraging relationships between source and target classes but still require prior knowledge about the target labels through annotated samples from the target domain. These challenges highlight the need for a more flexible approach that can simultaneously manage domain shifts while accommodating new taxonomies *without requiring additional annotations from the target domain*. Despite the significance of this problem, fully unsupervised methods that can address both domain shifts and taxonomy discrepancies remain largely underexplored.

Recent advancements in foundation models offer a promising direction to overcome these limitations. By leveraging large-scale pretraining on diverse datasets, these models can generalize across varied domains, tackling the challenges posed by limited domain knowledge and unseen categories. For instance, recent open-vocabulary semantic segmentation works (Ghiasi et al., 2022) leverage the knowledge in foundation models like CLIP (Radford et al., 2021) and Segment Anything (SAM) (Kirillov et al., 2023), enabling segmentation across a broad range of unbounded class labels. However, despite their strong generalization ability, foundation models often struggle with inferior performance compared to domain-specific models trained on in-domain datasets. Their broad focus can limit their effectiveness in specialized tasks, such as street scene understanding, where fine-grained segmentation requires detailed, domain-specific knowledge.

To conclude, existing works have been tackling image-level and label-level domain gaps separately, while the intersection of these two problems remains underexplored. In this work, we propose a new benchmark of unsupervised taxonomy adaptation, addressing both image-level and label-level domain gaps without supervision, as shown in Figure 1 (c). To achieve this, we propose DynAlign , a novel approach that integrates both domain-specific knowledge and rich open-world prior knowledge from foundation models. The framework first leverages domain-specific knowledge by aligning the data distributions of the source and target domains within the UDA paradigm. Subsequently, to incorporate prior knowledge from foundation models, DynAlign dynamically adapts to new scene contexts by retrieving in-domain predictions and extending the knowledge with foundation models to generate predictions in the new target label space, accommodating the different taxonomy in the target domain. In DynAlign , foundational model knowledge is fused in three modules. First, Large Language Model (LLM) is used for semantic taxonomy mapping and context-aware descriptions. For instance, the source domain label *'road'* is mapped to {*'road'*, *'sidewalk'*, *'lane marking'*, *etc.*} in the target domain, and then each label such as *'lane marking'* can be further enriched with more precise descriptions like {*'traffic lane marking'*, *'double lines'*} to capture the current context and improve semantic granularity. Then, SAM is employed to refine the coarse semantic masks generated by the UDA model, providing more fine-grained masks within precise boundaries (e.g., segmenting *'lane marking'*, *'catch basin'* within the *'road'* region). Lastly, a knowledge fusion mechanism is introduced, where CLIP is leveraged to extract textual features based on the context-aware taxonomy provided by LLM and reassign semantic labels to the proposed mask regions, effectively

fusing UDA knowledge with prior knowledge. Predictions from DynAlign can be directly used or leveraged as pseudo-labels to train an offline segmentation model on the target domain, enabling instant inference without the need for target domain annotations.

Our approach operates in a fully unsupervised manner, integrating seamlessly with any UDA-based semantic segmentation model. By leveraging the prior knowledge of foundation models, it strengthens in-domain predictions and flexibly adapts to new classes and scene contexts. When the target label set changes, only the taxonomy mapping needs to be updated to instantly predict on the target dataset without requiring additional training. This adaptability offers a highly flexible solution for cross-domain semantic segmentation in dynamic, real-world environments with changing taxonomies. We conduct extensive experiments on the GTA→ Mapillary Vistas and GTA → IDD street scene semantic segmentation benchmarks. The results demonstrate that our method effectively combines domain-specific knowledge with prior knowledge from foundation models, achieving superior performance in the unsupervised taxonomy-adaptive domain adaptation task.

## 2 RELATED WORKS

### 2.1 UNSUPERVISED DOMAIN ADAPTATION

Unsupervised domain adaptation (UDA) aims to minimize the domain gap and transfer knowledge from a labeled source domain to an unlabeled target domain. Due to the ubiquity of domain gaps, UDA methods have been widely applied to major computer vision problems including image and video classification (Zhang et al., 2023; Lai et al., 2023; Zara et al., 2023), object detection (Chen et al., 2018; Li et al., 2022b;c; Fan et al., 2023b), and semantic segmentation (Tsai et al., 2018; Hoyer et al., 2022a; 2023). UDA approaches typically minimize domain gaps through methods like discrepancy minimization (Long et al., 2015), adversarial training (Ganin et al., 2016; Long et al., 2018; Shi & Liu, 2024), self-training (Pan et al., 2019; Mei et al., 2020; Zhang et al., 2021). Recently, foundation models have been used to further enhance the adaptation performance by leveraging large-scale pretraining (Fahes et al., 2023; Tang et al., 2024; Gondal et al., 2024).

While traditional UDA methods assume a consistent label space between the source and target domains , this assumption is often violated in real-world scenarios. To address this, several specialized UDA methods have been developed to handle different label shift scenarios (Tachet des Combes et al., 2020; Garg et al., 2023; Westfechtel et al., 2023). Partial DA (Cao et al., 2018; Guo et al., 2022) addresses situations where the target domain contains a subset of the source domain's classes. Open-set DA (Saito & Saenko, 2021; Li et al., 2023) handles cases where the target domain includes unknown classes not present in the source domain. Universal DA (You et al., 2019; Qu et al., 2024) aims to adapt to target domains with any combination of known and unknown classes.

Typically, In the field of cross-domain semantic segmentation, various UDA approaches have been proposed to address the challenges posed by the fine-grained task. Class-incremental DA (Kundu et al., 2020) focuses on adding new classes while preserving knowledge of previously learned ones. Open-set adaptation methods (Bucher et al., 2021; Choe et al., 2024) aim to predict the boundaries of unknown classes. However, these works can only distinguish between the unknown classes and the known ones and do not perform further classification on the unknown classes. Taxonomy adaptive DA (Gong et al., 2022; Fan et al., 2023a) utilizes a more flexible taxonomy mapping where the target label space differs from the source domain. Despite being able to distinguish novel target classes further, these methods cannot be applied in a fully unsupervised DA setting as they rely on few-shot labeled samples from the target domain to gain knowledge on novel classes. Instead, DynAlign leverage prior knowledge from foundation models for unsupervised adaptation.

### 2.2 OPEN-VOCABULARY SEMANTIC SEGMENTATION

Open-vocabulary semantic segmentation aims to assign a semantic label to each pixel of an image using an arbitrary open-vocabulary label set (Xian et al., 2019; Bucher et al., 2019). Recent advancements in vision-language models like CLIP (Radford et al., 2021) have enabled zero-shot classification with enhanced generalization ability from large-scale pretraining, leading to their wide application in open-vocabulary tasks, including semantic segmentation (Li et al., 2022a).

To adapt CLIP for better performance in dense prediction tasks, different approaches have been explored, including fine-tuning CLIP (Liang et al., 2023; Cho et al., 2024) and modifying CLIP's model architecture (Wang et al., 2023; Yu et al., 2024). For finer segmentation boundaries, two-stage frameworks involving mask proposal generation followed by classification remain prominent in open-vocabulary semantic segmentation (Kirillov et al., 2023; Wu et al., 2023a; Ding et al., 2022). The recent advent of Segment Anything Model (SAM) (Kirillov et al., 2023) provides a free and precise mask proposal generation approach and thus integrated by several works to enhance the performance (Shi & Yang, 2024; Li et al., 2024; Yuan et al., 2024). Despite these advancements, open-vocabulary methods struggle with domain-specific nuances and fine-grained categories (Zhou & Beyerer, 2023; Wu et al., 2023b). Wei et al. (2024) leverages vision foundation models to improve the generalization ability of task-specific semantic segmentation models. Yilmaz et al. (2024) incorporates domain-specific knowledge into the open-vocabulary framework through supervised prompt fine-tuning, whereas our work focuses on fully unsupervised adaptation.

## 3 PROBLEM DEFINITION

We formulate the problem of unsupervised taxonomy adaptive cross-domain semantic segmentation as follows: given a labeled source domain $\mathcal{D}_s$ with a label set $\mathbb{C}_s$, our goal is to achieve semantic segmentation on an unlabeled target domain $\mathcal{D}_t$ with a known label set $\mathbb{C}_t$. Specifically, we have:

- a labeled source domain $\mathcal{D}_s = \{X_s, Y_s\}$, where $X_s \in \mathbb{R}^{H \times W \times 3}$ represents RGB images and $Y_s$ denotes pixel-wise annotations in the source label set $\mathcal{C}_s = \{c_s^1, c_s^2, \ldots, c_s^m\}$.
- an unlabeled target domain $\mathcal{D}_t = \{X_t\}$. The ground truth pixel-wise annotations belong to the known target label space $\mathbb{C}_t = \{c_t^1, c_t^2, ..., c_t^n\}$ and are not available during training.
- $\mathbb{C}_s$ and $\mathbb{C}_t$ may have inconsistent taxonomies. This inconsistency may involve differences in label granularity, hierarchical class structures (such as subclasses), or the introduction of entirely new categories in $\mathbb{C}_t$, as illustrated in Figure 1 (d).

Formally: Let $P_s$ and $P_t$ represent the distributions of source domain data $X_s$ and target domain data $X_t$, respectively. In the context of taxonomy-adaptive cross-domain semantic segmentation, three primary challenges need to be addressed:

- Image-level domain gap: the source and target data have distinct data distributions ($P_s \neq P_t$).
- Label-level taxonomy inconsistency: the source and target label sets are different ($\mathbb{C}_s \neq \mathbb{C}_t$).
- Absence of target domain annotations: no labeled data is available for the target domain.

We aim to train a model using both the labeled source domain $\mathcal{D}_s$ and the unlabeled target domain $\mathcal{D}_t$, and evaluate the performance on the target dataset $\mathcal{D}_t$ within the label space $\mathbb{C}_t$.

## 4 METHOD

**Method Overview**  In this work, we propose DynAlign, a novel framework for unsupervised taxonomy-adaptive cross-domain semantic segmentation that fuses domain-specific knowledge with text and visual prior knowledge. Our method comprises three main stages: incorporating domain knowledge, incorporating prior knowledge, and combining them through a knowledge fusion mechanism, as illustrated in Figure 2. In the first stage, we integrate domain-specific knowledge by training a domain-specific model on the available labeled source domain and unlabeled target domain, which generates predictions within the source label space (Section 4.1). In the second stage, we incorporate both text & vision prior knowledge to address the label-level taxonomy inconsistency. Specifically, we use LLM to construct a taxonomy mapping to link the source labels with the target labels and further enrich the target labels with context descriptions (see Section 4.2). Then we utilize SAM to generate mask proposals that segment the image into fine-grained semantic regions and extract multi-scale visual information for each region (see Section 4.3). In the final stage, we introduce knowledge fusion mechanism (see Section 4.4), where the domain-specific predictions and prior knowledge are integrated based on CLIP. As shown in Figure 3, for each mask proposal from SAM, we take the majority pixel label of in-domain prediction as the initial source label and retrieve the correlated target classes from the taxonomy mapping. CLIP is then used to extract the multi-scale regional visual features and the context-aware text features for the mapped target classes. Finally, each fine-grained mask region is reclassified based on CLIP feature similarity.

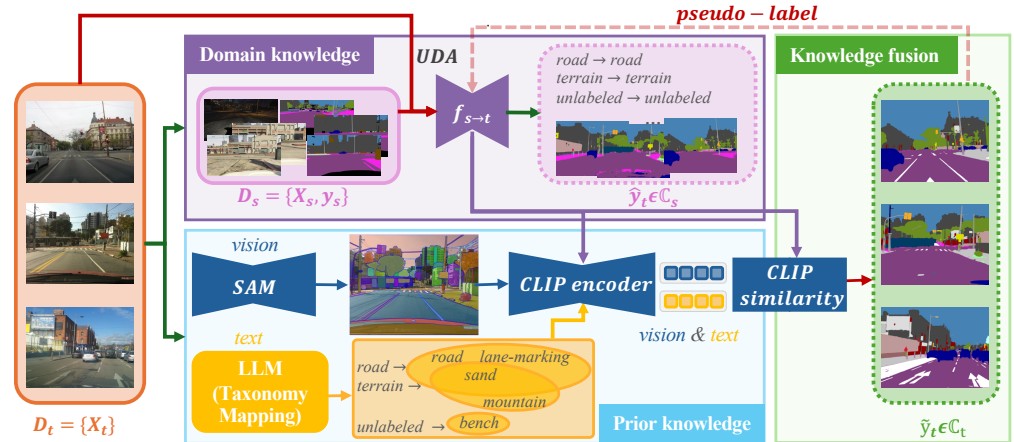

Figure 2: **DynAlign overview.** DynAlign integrates with any UDA model, leveraging its domain-specific knowledge and enhancing it with prior knowledge from foundation models. DynAlign starts with coarse UDA model predictions, followed by: 1) LLM constructing taxonomy mappings to align source and target domains; 2) SAM generating fine-grained masks. CLIP is deployed to fuse the visual knowledge from SAM with the semantic knowledge from LLM to reassign accurate labels. The CLIP-fused predictions can be used as pseudo-labels to further fine-tune the UDA model.

## 4.1 DOMAIN-SPECIFIC KNOWLEDGE

To gain domain-specific knowledge, given a labeled source domain $\mathcal{D}_s$ and an unlabeled target domain $\mathcal{D}_t$, we train a domain-specific model $f_{s \rightarrow t}$ by leveraging data from both domains. Specifically, we follow the UDA paradigm, which aligns the data distribution of the source and target domains. This adapted model integrates supervised knowledge from the source domain with visual information from the target domain, enabling it to generate accurate predictions on the unlabeled target dataset $\mathcal{D}_t$ within the source label space $\mathbb{C}_s$.

In our experiments, we develop an UDA model inspired by the architecture proposed by Hoyer et al. (2022b). The model comprises a hierarchical transformer encoder, based on the design of Xie et al. (2021), combined with a multi-scale decoder that effectively integrates contextual information from low-level features. Initially, the model is trained using a supervised cross-entropy loss on the labeled source domain $\mathcal{D}_s$. Then, we adapt it to the unlabeled target domain $\mathcal{D}_t$ through an unsupervised self-training. This adaptation process incorporates a teacher network that generates pseudo-labels for the target domain, which are then weighted based on confidence estimates to account for uncertainty. These weighted pseudo-labels are used to further refine the model's performance on the target domain. The adapted model learns shared feature representations for both domains, enabling it to effectively handle image-level domain shifts. Detailed model architecture and training procedures are provided in the Appendix A.1. We leverage the knowledge learned by $f_{s \rightarrow t}$ to generate initial predictions $\hat{y}_t$ for the target images within the source label set $\mathbb{C}_s$.

## 4.2 SEMANTIC TAXONOMY MAPPING

The acquired domain-specific model is constrained by the in-domain knowledge, thus limiting their ability to generalize beyond the learned source label space to a new target label space. To bridge this gap, we introduce a taxonomy reasoning process that adapts the label space from $\mathbb{C}_s$ to $\mathbb{C}_t$. This taxonomy mapping enables the model to semantically link source domain labels to the known target domain label set $\mathbb{C}_t$. For instance, the source label *'road'* can be mapped to more granular target domain labels such as {*'road', 'sidewalk', 'curb', 'lane marking', 'rail track', etc.*}. We construct a taxonomy mapping for each source label to connect the source and target label spaces. This mapping is flexibly defined, allowing for differences in label granularity, hierarchical class structures (such as subclasses), or the introduction of entirely new categories, as shown in Figure 1 (d).

Formally, given the source domain label set $\mathbb{C}_s = \{c_s^1, c_s^2, \ldots, c_s^m\}$ and the target domain label set $\mathbb{C}_t = \{c_t^1, c_t^2, ..., c_t^n\}$, we define the taxonomy mapping for each source domain label $c_s^i$ as:

$$c_s^i \rightarrow \mathbb{C}_t^i \subseteq \mathbb{C}_t, \quad 1 \leq i \leq m \tag{1}$$

where $\mathbb{C}_t^i$ represents an arbitrary subset of $\mathbb{C}_t$ that semantically correlates with the source domain label $c_s^i$. For novel classes that are not present in the source domain, we map them to each source

label, enabling better discovery of these new classes. Further details about the taxonomy mapping can be found in the Appendix A.7.

To extract meaningful semantic features, we use the CLIP text encoder to encode text features for each target label. Original class labels often lack sufficient context and semantic richness, leading to ambiguity and imprecision. For example, the label *'bridge'* in street scene segmentation datasets is too generic and might be interpreted as a bridge over a river rather than a pedestrian bridge in urban environments. This lack of specificity can cause confusion and misclassification in the segmentation process, resulting in inferior performance. To address this, we enhance the semantic clarity of each target class label by expanding it with contextually relevant synonyms or related phrases. For each target class label, we use GPT-4 (Achiam et al., 2023) to generate a set of contextually relevant terms, such as *'bridge'* → {*'road bridge', 'footbridge', 'pedestrian bridge', etc.*}. These terms are generated by prompting GPT-4 with descriptions of the dataset context and the original class label. More details are provided in the appendix A.8. This additional contextual information helps reduce ambiguity and improve the accuracy of mask region classification. We then compute the **context-aware text feature** for each target domain label. Specifically, for each target domain label $c_t^j \in \mathbb{C}_t$, we generate a context description set $\mathbb{C}_{context}^j$ and compute the CLIP encoded text feature for each description $c_{context} \in \mathbb{C}_{context}^j$. The text feature for class label $c_t^j$ is then obtained by averaging all the contextual description features, as follows:

$$F_t^j = \text{average } \Phi^T(c_{context}), \quad c_{context} \in \mathbb{C}_{context}^j \tag{2}$$

where $\Phi^T(\cdot)$ represents CLIP text encoder. We then aggregate these features to form the final text feature representations of the target domain label space denoted as $\mathbb{F}_{target} = \{F_t^1, F_t^2, ..., F_t^n\}$. For each source label $c_s^i$, we retrieve its mapped target labels $\mathbb{C}_t^i$ and the corresponding feature representation set $\mathbb{F}_T^i \subseteq \mathbb{F}_{target}$ as the semantic representation.

## 4.3 VISUAL PRIOR KNOWLEDGE

To identify new labels in the target domain, it is essential to align visual information with the target domain label set and its novel semantic categories. Therefore, we incorporate prior knowledge from general-purpose foundation models to enhance visual understanding. We first generate mask proposals using the Segment-Anything-Model (SAM) (Kirillov et al., 2023), which is renowned for its zero-shot capability to produce high-quality, fine-grained masks with precise segmentation boundaries. With SAM, we are able to obtain a set of mask regions that likely correspond to semantically meaningful object boundaries.

Next, we capture semantic visual information for each mask region by extracting visual embeddings using CLIP, which requires capturing both fine-grained details and broader contextual information. To obtain more representative features, we propose **multi-scale visual feature extraction**, which concatenates local and multi-scale global features. Given a target domain image $x_t$, and a binary mask proposal $m$ from SAM, we obtain the masked local region $r = x_t \odot m$ and a bounding box $b$ that crops the masked region. The local feature is extracted with CLIP vision encoder $\Phi^V(\cdot)$, as:

$$F_l = \Phi^V(b) \tag{3}$$

Specifically, we use ConvCLIP (Yu et al., 2024) vision encoder due to its advantage in dense prediction tasks. To capture broader contextual information, we incorporate global features by averaging embeddings across multiple scales. This is achieved by adding multi-scale padding around each local bounding box region $b$, as illustrated in Figure 3. The padding size is adjusted based on the class labels, with larger objects like *'road'* assigned larger padding sizes, and smaller objects like *'bicycle'* assigned smaller padding sizes. For each bounding box $b$, we create a set of padded global regions $\mathbb{B}$ and extract the corresponding global feature:

$$\mathbb{F}_g = \{\Phi^V(b_k)\}, \quad b_k \in \mathbb{B} \tag{4}$$

To combine both local and global visual information, we extract the final visual feature $F_V$ as the weighted sum of $\mathbb{F}_g$ and $F_l$. Specifically, we calculate the cosine similarity between the local feature

$F_l$ and each global feature $F_{g_k} \in \mathbb{F}_g$. The similarity scores $\phi_k$ are computed as follows:

$$\phi_k = \langle F_l, F_{g_k} \rangle = \frac{(F_l)^T F_{g_k}}{\|F_l\| \|F_{g_k}\| + \epsilon}, \quad F_{g_k} \in \mathbb{F}_g. \tag{5}$$

where $\epsilon$ is a small constant. Subsequently, we derive the final multi-scale visual feature $F_V$ for the current mask region by aggregating the local feature $F_l$ with the global features $F_g$, weighted by their respective cosine similarities $\phi_k$. This aggregation ensures that both local details and broader contextual information contribute effectively to the visual feature representation:

$$F_V = \frac{\Sigma(\phi_k F_{g_k} + (1 - \phi_k)F_l)}{|\mathbb{F}_g|}, \quad F_{g_k} \in \mathbb{F}_g. \tag{6}$$

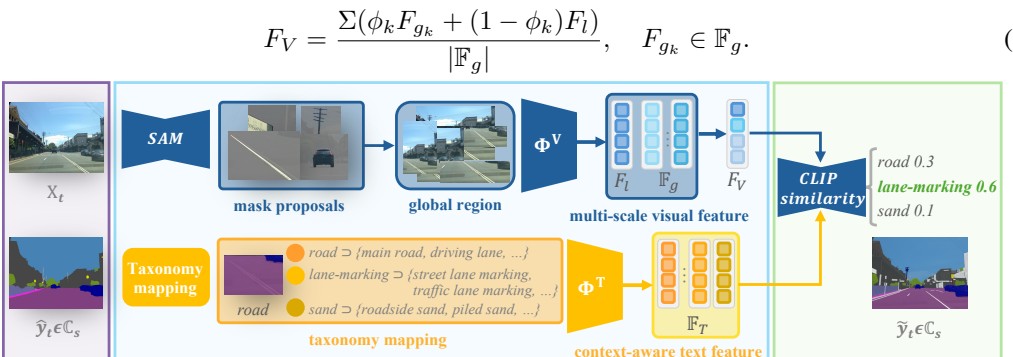

Figure 3: **Foundational models and knowledge fusion.** The fine-grained mask proposals from SAM are encoded into multi-scale visual features using CLIP's vision encoder, while the enriched target domain taxonomies from LLM are encoded as context-aware text features via CLIP's text encoder. The similarity between these visual and text embeddings is then calculated to reassign semantic taxonomies accurately to the fine-grained masks in the target domain. Here, $\Phi^V(\cdot)$ and $\Phi^T(\cdot)$ denote the CLIP vision and text encoders, respectively. $F_l$ and $\mathbb{F}_g$ represent the local and global features, while $F_V$ denotes their weighted sum, forming the final extracted multi-scale visual feature to represent the mask region. $\mathbb{F}_T$ refers to the extracted text feature set of candidate classes.

## 4.4 KNOWLEDGE FUSION

Given the domain-specific knowledge and prior knowledge from foundation models, we aim to fuse them to bridge both image-level and label-level domain gaps. The general fusion process of DynAlign is shown in Figure 3.

Given a target image $x_t$, we generate mask proposals that segment the image into fine-grained regions using SAM. For each mask region $m$, we first associate it with the source domain label space by retrieving in-domain knowledge from the label prediction $\hat{y}_t$ generated by model $f_{s \to t}$ (as described in Section 4.1) The initial label $c_s^i$ for mask region $m$ is determined by the majority of predicted pixel labels within that mask area. To reassign each mask region with new target labels, given the source label $c_s^i$, we retrieve its related target domain label subsets $c_s^i \to \mathbb{C}_t^i \subseteq \mathbb{C}_t$ and the corresponding feature representation set $\mathbb{F}_T \subseteq \mathbb{F}_{target}$ using prior knowledge from the taxonomy mapping (see Section 4.2). Additionally, based on the initial label $c_s^i$, we adjust the appropriate padding size of the global region $\mathbb{B}$ (see Section 4.3) accordingly to enrich the visual context and extract the multi-scale visual feature $F_V$, serving as the regional visual feature representation. Finally, each mask region is reassigned a new target label based on the largest similarity between the context-aware text features $\mathbb{F}_T$ and multi-scale visual feature $F_V$ as:

$$\phi = \langle F_V, \mathbb{F}_T \rangle, \quad \tilde{y}_t = \operatorname{argmax}(\phi) \tag{7}$$

We reassign the label of each mask region from $\hat{y}_s$ to the newly estimated $\tilde{y}_t$ when the confidence exceeds 0.5. This reassignment is applied to every mask proposal within the image sample $x_t$, thereby updating the pixel-wise semantic pseudo-labels in the target domain.

DynAlign generates predictions for new classes in the target domain that were not present in the source domain, eliminating the need for manual annotations. This capability allows the model to adapt flexibly to new target classes and when the target label set changes, we simply redefine the

taxonomy mapping to generate new predictions for the target domain dataset. This two-step process–mapping known classes and discovering new ones – enables our framework to effectively adapt to the target domain's taxonomy in a fully unsupervised manner.

# 5 EXPERIMENT

## 5.1 IMPLEMENTATION DETAILS

**Datasets** We evaluate our method using the synthetic dataset GTA as the source domain and two real-world datasets, Mapillary Vistas and India Driving Dataset (IDD), as target domains. **GTA** (Richter et al., 2016) is a synthetic dataset with pixel-level annotations for 19 semantic classes. It serves as the labeled source domain in our experiments. **Mapillary Vistas** (Neuhold et al., 2017) contains 25k high-resolution images collected from a wide variety of environments, weather conditions, and geographical locations. It is annotated with 66 object categories, providing a highly challenging real-world target domain. **IDD** dataset (Varma et al., 2019), captured from Indian urban driving scenes, is characterized by complex and varied road environments. It contains 10,003 images annotated with 25 semantic classes at level 3 granularity. For evaluation, we focus on 45 classes from Mapillary Vistas and 24 from IDD, excluding small-scale or less informative categories.

**Experimental setup** For the unsupervised taxonomy-adaptive DA task, we use the GTA training set as the labeled source domain and the training sets of Mapillary Vistas and IDD as the unlabeled target domains. No target domain annotations are used during training, ensuring a fully unsupervised domain adaptation setup. We perform image-level adaptation by training the model $f_{s \rightarrow t}$ on the labeled source domain and each of the unlabeled target domains separately, following the default training parameters in Hoyer et al. (2022b). For label-level adaptation, we define a taxonomy mapping between the source and target labels (see Appendix A.7) and context descriptions for target labels (see Appendix A.8). Performance is reported on the validation set of each target dataset. We employ the ViT-L SAM model (Kirillov et al., 2023) and the ConvNeXt-Large CLIP model (Yu et al., 2024) by default. More experimental details are provided in the appendix A.1.

**Evaluation metrics** We evaluate the performance using two standard semantic segmentation metrics. Mean Intersection over Union (mIoU) calculates the average intersection over union for each class and then averages across all classes. Mean Accuracy (mAcc) calculates the percentage of correctly predicted pixels for each class and then averages the results across all classes.

## 5.2 EXPERIMENTAL RESULTS

To the best of our knowledge, our work is the first to investigate the fully unsupervised taxonomy-adaptive domain adaptation problem. Traditional DA methods are limited to fixed label spaces, making them inapplicable to our problem setting. Therefore, we compare our proposed method, DynAlign , with open-vocabulary segmentation methods, Grounded-SAM (Ren et al., 2024) and OWL-VIT (Matthias Minderer, 2023). Additionally, we evaluate the naive combination of these methods with HRDA, where HRDA generates predictions for classes matching the source labels, while open-vocabulary methods are employed to predict new classes (see Appendix A.3 for more illustration). The comparative results are summarized in Table 1.

Table 1: Open-vocabulary semantic segmentation comparisons on Mapillary Vistas and IDD

| Methods | Mapillary Vistas | | | | | | IDD | | | | | |
| | all | | known | | unknown | | all | | known | | unknown | |
| | mACC | mIoU | mACC | mIoU | mACC | mIoU | mACC | mIoU | mACC | mIoU | mACC | mIoU |
|---|---|---|---|---|---|---|---|---|---|---|---|---|
| Grounded-SAM | 33.1 | 28.6 | 46.7 | 43.0 | 24.0 | 18.3 | 36.0 | 30.8 | 43.7 | 37.2 | 16.1 | 14.1 |
| OwlVIT-SAM | 29.6 | 19.5 | 32.7 | 26.7 | 27.5 | 14.7 | 31.6 | 20.9 | 33.3 | 23.1 | 27.3 | 15.2 |
| HRDA | - | - | 75.8 | 65.8 | - | - | - | - | 68.9 | 61.3 | - | - |
| HRDA + Grounded-SAM | 40.4 | 32.9 | 63.6 | 53.1 | 25.0 | 19.4 | 51.7 | 39.4 | 65.6 | 49.3 | 16.1 | 14.1 |
| HRDA + OwlVIT-SAM | 40.2 | 28.8 | 56.1 | 48.4 | 29.5 | 15.6 | 55.9 | 40.2 | **67.1** | 49.9 | 27.3 | 15.2 |
| Ours | **53.0** | **36.7** | **72.6** | **62.4** | **39.9** | **19.6** | **57.7** | **41.7** | 66.8 | **50.9** | **34.3** | **18.1** |

On the Mapillary dataset, open-vocabulary methods, which relies solely on foundation models, demonstrate limited performance. By integrating domain-specific knowledge through HRDA,

the performance improves, with HRDA + Grounded-SAM reaching 32.9% and HRDA + OWL-ViT-SAM achieving 28.8% mIoU. Notably, our method, DynAlign , significantly outperforms all baselines, achieving the highest mIoU of 36.7%. Results on IDD show consistent trends. Open-vocabulary methods alone perform modestly with mIoUs of 30.8% for Grounded-SAM and 20.9% for OWL-ViT-SAM. When combined with HRDA, performance improves to 39.4% and 40.2%, respectively. DynAlign demonstrates again superior performance with the highest mIoU of 41.7%. Notably, for unknown classes, DynAlign achieves an mIoU of 18.1%, surpassing HRDA + Grounded-SAM at 14.1% and HRDA + OWL-ViT-SAM at 15.2%, showing a substantial improvement in accurately segmenting unseen categories. The results demonstrate that incorporating domain-specific knowledge through UDA significantly enhances performance, even when leveraging strong foundation models for open-vocabulary tasks. DynAlign not only preserves the domain knowledge from the UDA baseline but also retains the flexibility to accommodate new classes. This highlights its advantages in effectively integrating both domain knowledge and prior knowledge, highlighting its advantage over traditional UDA methods and open-vocabulary approaches.

We present qualitative comparison results on the Mapillary Vistas dataset in Figure 5. The key new classes are highlighted under each set of results. Our method clearly outperforms the baselines by producing more precise boundaries, better detection of new classes, and more accurate predictions.

## 5.3 PSEUDO-LABEL TRAINING FOR SEGMENTATION MODEL

While DynAlign demonstrated superior performance over baseline open-vocabulary methods during inference, we also explored its capability to generate pseudo-labels for training a new segmentation model on the target domain's taxonomy. This approach aims to reduce inference time and potentially boost segmentation accuracy. By generating pseudo-labels on the Mapillary Vistas training set using DynAlign , we train a semantic segmentation model in the new label space (details provided in Appendix A.1). Figure 4 compares direct inference using DynAlign and performance of the trained segmentation model on the validation set. The pseudo-label trained model outperforms direct inference, with mACC improving from 53.0% to 56.2% and mIoU from 36.7% to 38.9%. These results suggest that DynAlign can generate high-quality pseudo-labels, en-

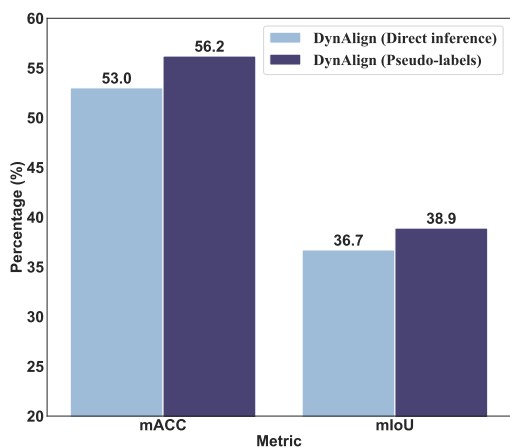

Figure 4: Performance comparison between direct inference and pseudo-label training using DynAlign on the Mapillary Vistas dataset.

abling the training of segmentation models that can greatly boost the efficiency and maintain improved performance on the target domain. We report the memory usage and computational efficiency of DynAlign in both direct inference and pseudo-labeling in the appendix A.2.

Table 2: Ablations on multi-scale visual feature and context-aware text feature

| Modules | | Mapillary Vistas | | IDD | |
|---|---|---|---|---|---|
| MS (vision) | CA (text) | mACC | mIoU | mACC | mIoU |
| ✗ | ✗ | 44.0 | 29.0 | 52.9 | 37.9 |
| ✗ | ✓ | 45.4 | 29.4 | 52.5 | 38.0 |
| ✓ | ✗ | 50.2 | 35.6 | 57.0 | 39.9 |
| ✓ | ✓ | **53.0** | **36.7** | **57.7** | **41.7** |

Table 3: Ablations on CLIP vision encoders

| model | Mapillary Vistas | | IDD | |
|---|---|---|---|---|
| | mACC | mIoU | mACC | mIoU |
| CLIP | 50.8 | 35.0 | 54.7 | 38.0 |
| MaskCLIP | 50.5 | 34.7 | 54.1 | 37.5 |
| SCLIP | 50.9 | 36.0 | 56.8 | 39.4 |
| ConvCLIP | **53.0** | **36.7** | **57.7** | **41.7** |

## 5.4 ABLATION STUDIES

**Multi-scale visual feature & context-aware text feature ablation**  In contrast to the basic CLIP features, we enhance feature representations by introducing **context-aware text feature (CA-Text)** and **multi-scale visual feature (MS-vision)**. As shown in Table 2, these two strategies have distinct impacts on the performance. Incorporating multi-scale visual features improves the Mapillary mIoU

from 29.0% to 35.6% and the IDD mIoU from 37.9% to 39.9%. The context-aware text feature shows modest gains, particularly on Mapillary, where mIoU rises from 29.0% to 29.4%. Combining both components yields the best results (Mapillary mIoU: 36.7%, IDD mIoU: 41.7%).

**DynAlign modules.** Table 3 compares the performance of various CLIP backbones in DynAlign framework. ConvCLIP (Yu et al., 2024), the only convolution-based model, outperforms ViT-based models including CLIP (Radford et al., 2021), MaskCLIP (Dong et al., 2023), and SCLIP (Wang et al., 2023), and achieves the best results, with a Mapillary Vistas mIoU of 36.7% and an IDD mIoU of 41.7%. The result shows its advantage in dense prediction tasks when input sizes scale up.

Table 4 shows shows the performance on the Mapillary Vistas dataset when replacing corresponding modules in DynAlign with MIC (Hoyer et al., 2023), Mobile-SAM (Zhang et al.), and Llama (Touvron et al., 2023). The results demonstrate consistently strong performance across various model integrations, emphasizing DynAlign 's flexibility in seamlessly incorporating foundation models into domain-specific tasks.

Table 4: Ablations on DynAlign modules

| Method | mIoU | mAcc |
|---|---|---|
| HRDA → MIC | 37.7 | 54.4 |
| SAM → MobileSAM | 35.2 | 51.5 |
| GPT-4 → Llama | 37.5 | 52.1 |
| DynAlign (Ours) | 36.7 | 53.0 |

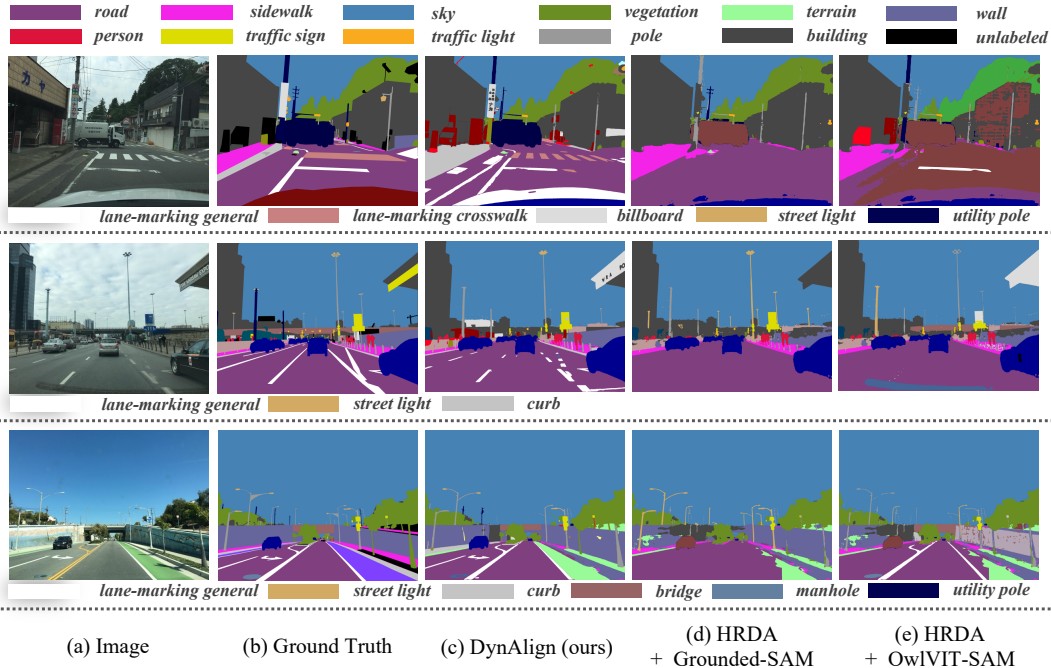

Figure 5: **Qualitative comparisons on Mapillary Vistas dataset.** DynAlign effectively segments new and fine-grained classes on the target domain, showing strong taxonomy adaptation capabilities.

## 6 CONCLUSION

In this paper, we propose DynAlign to address the challenge of unsupervised taxonomy-adaptive cross-domain semantic segmentation, effectively segmenting images across domains with different taxonomies without requiring target domain annotations. DynAlign integrates domain-specific knowledge by utilizing UDA models to bridge the image-level domain gap, and leverages foundation models to resolve label-level taxonomy inconsistencies between domains. The approach demonstrates significant improvements over existing methods on the Mapillary Vistas and IDD street scene datasets, consistently achieving higher mIoU scores for both known and unknown classes.

To the best of our knowledge, we are the first to define and address the unsupervised taxonomy-adaptive domain adaptation problem. Our results demonstrate that DynAlign outperforms not only open-vocabulary segmentation methods but also their naive combinations with domain adaptation techniques. While our research focuses on road scene understanding, the framework has the potential to be extended to other domains with evolving taxonomies.

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

# A  APPENDIX

## OVERVIEW

The supplementary material presents the following sections to strengthen the main manuscript:

— **Sec. A.1** includes more implementation details.

— **Sec. A.2** presents memory usage and computational efficiency comparison.

— **Sec. A.3** provides examples and further explanation of naive combination of HRDA with open-vocabulary approaches, as mentioned in Table 1

— **Sec. A.4** presents ablations on CLIP confidence thresholding.

— **Sec. A.5** provides additional experimental results under different DA settings.

— **Sec. A.6** provides detailed per-class experimental results in Table 1.

— **Sec. A.7 and Sec. A.8** provides details of building taxonomy mapping and context descriptions for each target label.

## A.1  MODEL ARCHITECTURE AND TRAINING

**HRDA for UDA segmentation model** For the UDA segmentation model, we follow HRDA (Hoyer et al., 2022b). We follow the HRDA framework, which combines large low-resolution (LR) crops for capturing scene context with small high-resolution (HR) crops for fine detail. These two types of inputs are combined using a learned scale attention mechanism, which enables the model to effectively handle multi-resolution information. In addition, HRDA uses overlapping slide inference to refine the generated pseudo-labels, ensuring that the context and details of the images are well-represented in the segmentation maps. The HRDA framework builds on the DAFormer (Hoyer et al., 2022a) architecture, which incorporates a domain-robust transformer backbone to extract features and perform segmentation. A self-training strategy with a teacher-student model is used, where the teacher generates pseudo-labels for the target domain, and these pseudo-labels are weighted based on a confidence score to prevent error accumulation during training. The teacher model is updated using an exponential moving average (EMA) of the student model's weights, ensuring stable pseudo-labels over time. In our experiments, we follow the default training parameters in the GTA $\rightarrow$ CityScapes setting and adapt from the labeled GTA dataset to the unlabeled Mapillary Vistas/IDD respectively. The labeled GTA training set and unlabeled Mapillary Vistas/IDD training set are used to train the UDA Framework.

**Mask2Former for target domain segmentation** To train a domain-specific model on the target domain, we utilize the Mask2Former architecture. The model comprises a backbone network that extracts low-resolution features, a pixel decoder that upscales these features, and a transformer decoder for processing object queries. It deploys masked attention, which focuses on local regions of the predicted mask, improving convergence and handling small objects. In our experiment, we generate pseudo-labels on the target training set using DynAlign and train Mask2Former with these labels on the target dataset to improve segmentation performance.

All experiments are conducted on NVIDIA A100-SXM4-80GB GPU.

## A.2  COMPUTATIONAL EFFICIENCY

Our method comprises three main components: the UDA semantic segmentation model (HRDA), SAM, and CLIP. For semantic segmentation on high-resolution images, the memory usage primarily depends on the semantic segmentation model (HRDA), while the inference time is influenced by the reassignment of novel class labels. We provide detailed information on memory usage, inference time, and model parameters in table 5.

The total model parameter count of our method includes the UDA model parameters, plus 308M for SAM (ViT-L) and 351M for CLIP (ConvNeXt-Large). The reported total memory usage corresponds to the memory allocated during the complete inference process for a single image. Overall, our

Table 5: Computational efficiency and memory usage of DynAlign

| Model | Memory Usage | Inference Time | Model Parameters |
|---|---|---|---|
| HRDA | 23.9 GB | 1.9 s | 86 M |
| Ours (HRDA) | 34.8 GB | 64.1 s | 668 M |
| DAFormer | 10.5 GB | 0.7 s | 86 M |
| Ours (DAFormer) | 21.7 GB | 64.1 s | 668 M |
| Ours (pseudo-label, HRDA) | 23.9 GB | 1.9 s | 86 M |
| Ours (pseudo-label, DAFormer) | 10.5 GB | 0.7 s | 86 M |
| Ours (pseudo-label, Mask2Former) | 3.7 GB | 0.3 s | 215 M |

framework requires reasonable computational resources. Moreover, the UDA framework can be flexibly substituted to accommodate memory limitations. As shown in the table, replacing HRDA with DAFormer significantly reduces memory consumption. For efficient inference, a segmentation model can also be trained using pseudo-labels generated by our framework, as described in *Section 5.3*. For example, a trained Mask2Former model using our inference pseudo labels, indicated as Ours (pseudo-label, Mask2Former), achieves inference in just 0.3 seconds per single image, offering a significant efficiency boost while maintaining high performance.

### A.3 EXAMPLES OF NAIVE COMBINATION OF HRDA WITH OPEN-VOCABULARY APPROACHES

For a naive combination of in-domain predictions with prior knowledge as baseline methods, we take the in-domain segmentation results produced by HRDA and layer them with the new class predictions from open-vocabulary models. As demonstrated in Figure 6, the open-vocabulary predictions for known classes in the source domain label space are discarded. This ensures that only novel class predictions from the open-vocabulary model are added on top of the HRDA predictions, preserving in-domain knowledge from HRDA.

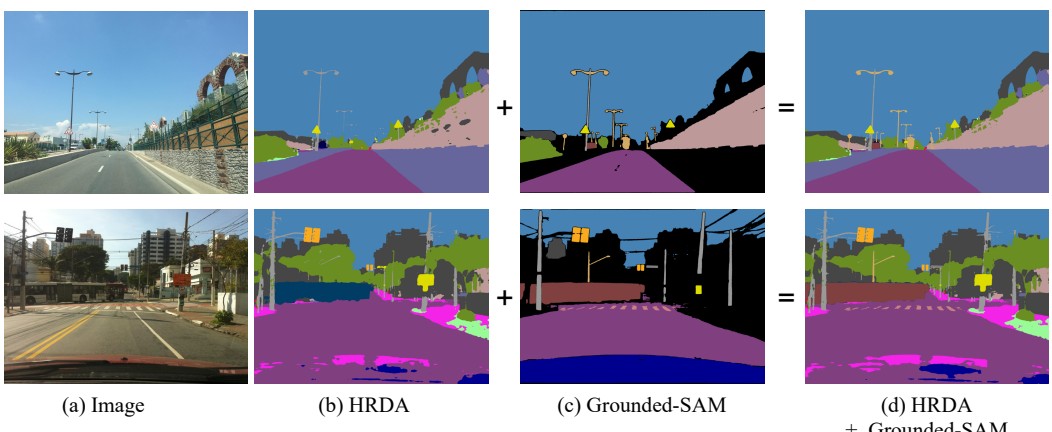

    (a) Image         (b) HRDA         (c) Grounded-SAM         (d) HRDA + Grounded-SAM

Figure 6: Illutration on HRDA+Grounded-SAM baseline

### A.4 ABLATIONS ON CLIP CONFIDENCE THRESHOLD

In our experiments, we set the confidence threshold to 0.5 by default for reassigning the class label. We conduct ablation studies with varying confidence thresholds, as presented in Table 6. The results show that while the mAcc improves with a lower confidence threshold for assigning new class labels, the mIoU may decrease correspondingly. Overall, DynAlign demonstrates robustness and is not very sensitive to the confidence threshold within a certain value range.

Table 6: Ablations on CLIP confidence threshold

| Threshold | 0.4 | 0.5 | 0.6 | 0.7 | 0.8 | 0.9 |
|---|---|---|---|---|---|---|
| mIoU | 35.9 | 36.7 | 37.6 | 36.5 | 36.6 | 36.5 |
| mAcc | 54.3 | 53.0 | 52.1 | 50.3 | 47.3 | 46.5 |

## A.5 EXPERIMENTS UNDER MORE DA SETTINGS

To validate the effectiveness of our methods in diverse practical scenarios, we implement additional experiments under the following settings: 1) Traditional UDA Setting: In this case, the target label space is identical to the source label space. We adapt from GTA to Cityscapes, both using the same label space. 2) Fine-to-Coarse Setting: Here, the target domain labels are coarser than the source domain labels. For this, we adapt from Mapillary (45 classes) to Cityscapes (19 classes). As shown in Table 7, in both the traditional UDA and fine-to-coarse settings, where sufficient domain knowledge about the label space is available, our framework preserves the in-domain performance and provides slight improvements when adequate in-domain knowledge is already present.

In Table 8, we present results under taxonomy-adaptive domain adaptation with a smaller label-space domain gap. Specifically, we adapt from Synthia (16 classes) to Cityscapes (19 classes). These results highlight that our method not only maintains strong performance on known classes but also predicts effectively for unknown classes.

Table 7: Semantic segmentation comparison on Mapillary under traditional UDA and fine-to-coarse setting

| Setting | mIoU | mAcc |
|---|---|---|
| GTA to Cityscapes (HRDA, 19 $\rightarrow$ 19) | 74.9 | 82.0 |
| GTA to Cityscapes (Ours, 19 $\rightarrow$ 19) | 75.9 | 83.7 |
| Mapillary to Cityscapes (Ours, 45 $\rightarrow$ 19) | 70.4 | 81.5 |

Table 8: Semantic segmentation comparison on Mapillary under coarse-to-fine setting with small label domain gap

| Settings | all classes | | known classes | | unknown classes | |
|---|---|---|---|---|---|---|
| | mIoU | mAcc | mIoU | mAcc | mIoU | mAcc |
| Synthia to Cityscapes (HRDA, 16 $\rightarrow$ 16) | N/A | N/A | 66.8 | 73.7 | N/A | N/A |
| Synthia to Cityscapes (Ours, 16 $\rightarrow$ 19) | 61.6 | 72.7 | 68.8 | 77.5 | 23.2 | 47.0 |

## A.6 PER-CLASS EVALUATION RESULTS

For experimental results in Table 1, we provide the corresponding per-class evaluation in Table 9 and Table 10.

## A.7 TAXONOMY MAPPING

We provide the detailed taxonomy mapping that maps each source label to its correlated target labels in Table 11 and Table 12. We utilizes GPT-4 to generate an initial proposal for potentially correlated taxonomy mappings between the source and target domains and introduce human intervention to refine these mappings due to the differing definitions of classes across datasets. For example, in the Mapillary dataset, the term "guard rail" specifically refers to roadside guard rails, whereas in other datasets or in general usage, it may represent a broader guarding fence. For the same reason, we exclude labels with inherent ambiguity, which we refer to classes that are broad or highly context-dependent. (For example, the class "other vehicles" may encompass various types of vehicles that are not explicitly defined in the dataset, making precise labeling a challenge. Similarly, the class "ego vehicle" refers specifically to the vehicle equipped with the camera, which often appears only partially in the image.) We examine the generated taxonomy mapping to ensure that the mappings

Table 9: Per-class semantic segmentation evaluation results on Mapillary. (lane marking is refered as lm).

| | Grounded-SAM | | Grounded-SAM + HRDA | | OwlVIT-SAM | | OwlVIT-SAM + HRDA | | Ours | |
|---|---|---|---|---|---|---|---|---|---|---|
| | IoU | Acc | IoU | Acc | IoU | Acc | IoU | Acc | IoU | Acc |
| curb | 1.6 | 1.6 | 1.6 | 1.6 | 18.7 | 27.0 | 18.7 | 27.0 | 16.3 | 44.9 |
| fence | 30.3 | 33.3 | 40.0 | 48.1 | 18.9 | 22.3 | 32.9 | 37.8 | 45.6 | 53.2 |
| guard rail | 21.3 | 28.8 | 21.3 | 28.8 | 17.9 | 36.1 | 17.9 | 36.1 | 15.3 | 21.1 |
| wall | 1.3 | 1.4 | 35.5 | 55.2 | 11.4 | 19.3 | 30.4 | 41.8 | 39.3 | 59.0 |
| rail track | 10.7 | 12.0 | 10.7 | 12.0 | 2.2 | 4.3 | 2.2 | 4.3 | 17.0 | 18.1 |
| road | 81.9 | 93.7 | 79.0 | 93.5 | 54.0 | 58.6 | 58.5 | 65.9 | 70.2 | 77.1 |
| sidewalk | 33.0 | 34.6 | 42.7 | 64.0 | 32.2 | 43.0 | 40.3 | 53.7 | 27.8 | 33.3 |
| bridge | 41.7 | 46.8 | 41.7 | 46.8 | 30.7 | 43.6 | 30.7 | 43.6 | 20.8 | 23.3 |
| building | 70.1 | 81.6 | 80.3 | 94.5 | 44.0 | 52.8 | 71.8 | 81.1 | 76.5 | 85.6 |
| tunnel | 0.0 | 0.0 | 0.0 | 0.0 | 1.2 | 3.6 | 1.2 | 3.6 | 13.4 | 79.5 |
| person | 67.9 | 78.6 | 74.9 | 85.6 | 29.5 | 33.3 | 69.0 | 78.3 | 77.7 | 89.3 |
| bicyclist | 8.3 | 9.4 | 21.6 | 36.4 | 10.6 | 12.6 | 35.1 | 66.8 | 53.8 | 72.0 |
| motorcyclist | 13.9 | 32.1 | 13.9 | 32.1 | 9.9 | 11.7 | 9.9 | 11.7 | 51.6 | 58.3 |
| lm - crosswalk | 20.3 | 24.4 | 20.3 | 24.4 | 8.9 | 14.8 | 8.9 | 14.8 | 20.0 | 56.6 |
| lm - general | 1.6 | 1.6 | 1.6 | 1.6 | 12.4 | 19.8 | 12.4 | 19.8 | 18.0 | 41.8 |
| mountain | 29.1 | 32.9 | 29.1 | 32.9 | 16.1 | 37.8 | 16.1 | 37.8 | 43.8 | 56.5 |
| sand | 29.6 | 35.7 | 29.6 | 35.7 | 24.1 | 38.1 | 24.1 | 38.1 | 18.4 | 49.2 |
| sky | 91.7 | 92.9 | 94.8 | 99.0 | 69.0 | 70.5 | 73.2 | 76.1 | 95.1 | 97.8 |
| snow | 34.1 | 34.5 | 34.1 | 34.5 | 26.9 | 38.9 | 26.9 | 38.9 | 40.9 | 77.5 |
| terrain | 12.3 | 12.5 | 48.1 | 93.1 | 0.3 | 0.3 | 47.6 | 82.7 | 48.5 | 84.4 |
| vegetation | 28.0 | 28.3 | 81.2 | 85.3 | 19.1 | 20.3 | 72.3 | 75.5 | 82.0 | 85.7 |
| water | 81.1 | 81.7 | 81.1 | 81.7 | 72.2 | 87.9 | 72.2 | 87.9 | 47.3 | 56.0 |
| banner | 5.3 | 5.4 | 5.3 | 5.4 | 6.8 | 10.0 | 6.8 | 10.0 | 0.5 | 0.7 |
| bench | 44.3 | 48.7 | 44.3 | 48.7 | 23.7 | 46.4 | 23.7 | 46.4 | 1.2 | 78.9 |
| billboard | 12.9 | 15.1 | 12.9 | 15.1 | 15.3 | 32.1 | 15.3 | 32.1 | 28.1 | 55.8 |
| catch basin | 0.0 | 0.0 | 0.0 | 0.0 | 0.0 | 0.0 | 0.0 | 0.0 | 8.9 | 18.1 |
| manhole | 19.3 | 19.8 | 19.3 | 19.8 | 20.4 | 40.5 | 20.4 | 40.5 | 23.7 | 25.3 |
| phone booth | 12.8 | 26.6 | 12.8 | 26.6 | 1.9 | 35.7 | 1.9 | 35.7 | 0.8 | 57.8 |
| street light | 8.0 | 46.4 | 8.0 | 46.4 | 3.0 | 32.6 | 3.0 | 32.6 | 15.2 | 21.2 |
| pole | 22.4 | 24.9 | 33.0 | 43.6 | 15.5 | 16.6 | 29.1 | 37.2 | 41.9 | 57.6 |
| traffic sign frame | 0.4 | 0.9 | 0.4 | 0.9 | 3.2 | 7.6 | 3.2 | 7.6 | 0.1 | 0.1 |
| utility pole | 27.0 | 31.1 | 27.0 | 31.1 | 23.2 | 29.5 | 23.2 | 29.5 | 23.8 | 25.2 |
| traffic light | 60.3 | 64.5 | 56.4 | 65.1 | 26.2 | 28.1 | 53.5 | 61.0 | 59.8 | 73.0 |
| traffic sign (back) | 0.0 | 0.0 | 0.0 | 0.0 | 7.8 | 19.9 | 7.8 | 19.9 | 3.4 | 5.1 |
| traffic sign (front) | 30.7 | 32.5 | 40.4 | 43.1 | 29.2 | 32.0 | 45.8 | 48.1 | 61.8 | 66.8 |
| trash can | 48.5 | 49.8 | 48.5 | 49.8 | 28.0 | 42.9 | 28.0 | 42.9 | 15.0 | 19.4 |
| bicycle | 57.1 | 63.6 | 49.9 | 54.3 | 28.1 | 58.1 | 49.0 | 53.3 | 57.1 | 62.9 |
| boat | 40.0 | 62.9 | 40.0 | 62.9 | 12.4 | 63.8 | 12.4 | 63.8 | 7.4 | 12.8 |
| bus | 76.0 | 81.7 | 60.9 | 67.1 | 39.7 | 60.6 | 51.8 | 56.7 | 68.3 | 73.1 |
| car | 62.1 | 64.5 | 65.1 | 68.1 | 26.9 | 34.6 | 49.6 | 51.7 | 88.1 | 92.0 |
| caravan | 0.0 | 0.0 | 0.0 | 0.0 | 0.0 | 0.2 | 0.0 | 0.2 | 24.1 | 82.6 |
| motorcycle | 49.2 | 52.8 | 53.2 | 61.4 | 37.2 | 39.7 | 54.9 | 62.9 | 65.1 | 76.6 |
| on rails | 0.0 | 0.0 | 9.0 | 10.2 | 0.0 | 0.0 | 28.1 | 30.4 | 53.3 | 59.3 |
| trailer | 0.0 | 0.0 | 0.0 | 0.0 | 0.1 | 5.7 | 0.1 | 5.7 | 1.4 | 20.0 |
| truck | 0.0 | 0.0 | 11.5 | 13.5 | 0.0 | 0.0 | 14.3 | 16.5 | 65.0 | 80.4 |
| **Average** | 28.6 | 33.1 | 32.9 | 40.4 | 19.5 | 29.6 | 28.8 | 40.2 | 36.7 | 53.0 |

are accurate and contextually appropriate. The taxonomy mapping significantly reduces effort compared to tasks such as pixel-wise manual annotation and only needs to be defined once per dataset pair. We highlight the identical classes in source and target domains in blue.

## A.8 CONTEXT NAMES

We provide the context names used to describe the target names under the scene context in Table 13, 14, and 15. We generate those names by providing GPT-4 with the instruction:

Table 10: Per-class semantic segmentation evaluation results on IDD

| | Grounded-SAM | | Grounded-SAM + HRDA | | OwlVIT-SAM | | OwlVIT-SAM + HRDA | | Ours | |
|---|---|---|---|---|---|---|---|---|---|---|
| | IoU | Acc | IoU | Acc | IoU | Acc | IoU | Acc | IoU | Acc |
| road | 83.5 | 88.9 | 84.6 | 94.5 | 76.0 | 81.9 | 84.9 | 96.4 | 85.9 | 99.1 |
| drivable fallback | 17.2 | 23.9 | 17.2 | 23.9 | 0.0 | 0.0 | 0.0 | 0.0 | 2.9 | 3.4 |
| sidewalk | 15.8 | 18.8 | 12.8 | 68.9 | 3.6 | 17.7 | 15.6 | 74.6 | 17.0 | 72.2 |
| non-drivable fallback | 13.8 | 18.8 | 13.7 | 14.2 | 1.9 | 2.4 | 14.1 | 14.6 | 4.9 | 5.1 |
| person | 25.1 | 71.8 | 57.3 | 72.7 | 13.6 | 27.4 | 56.8 | 71.8 | 59.5 | 74.6 |
| rider | 0.2 | 0.2 | 64.4 | 78.4 | 0.6 | 0.6 | 63.5 | 77.2 | 67.7 | 80.3 |
| motorcycle | 57.4 | 62.9 | 64.4 | 77.4 | 34.2 | 36.4 | 64.1 | 77.0 | 67.2 | 80.1 |
| bicycle | 33.7 | 36.2 | 15.2 | 38.5 | 2.6 | 34.0 | 16.2 | 38.4 | 15.8 | 39.5 |
| autorickshaw | 0.0 | 0.0 | 0.0 | 0.0 | 34.7 | 46.0 | 34.7 | 46.0 | 36.5 | 81.4 |
| car | 84.5 | 88.9 | 63.8 | 96.8 | 22.8 | 23.1 | 69.0 | 96.0 | 70.2 | 80.9 |
| truck | 72.3 | 77.8 | 86.2 | 91.7 | 31.6 | 35.2 | 80.7 | 84.9 | 86.8 | 91.9 |
| bus | 74.8 | 81.6 | 68.4 | 71.4 | 42.8 | 47.8 | 63.6 | 66.1 | 68.8 | 71.9 |
| curb | 6.8 | 7.3 | 6.8 | 7.3 | 14.9 | 30.6 | 14.9 | 30.6 | 14.7 | 18.2 |
| wall | 4.7 | 4.7 | 40.8 | 66.5 | 8.0 | 8.9 | 40.8 | 59.8 | 41.9 | 67.6 |
| fence | 8.3 | 21.2 | 20.2 | 33.8 | 5.5 | 13.4 | 21.5 | 34.9 | 22.3 | 35.1 |
| guard rail | 8.2 | 13.0 | 8.2 | 13.0 | 9.3 | 28.5 | 9.3 | 28.5 | 13.9 | 22.1 |
| billboard | 22.3 | 23.7 | 22.3 | 23.7 | 16.1 | 25.8 | 16.1 | 25.8 | 43.9 | 60.7 |
| traffic sign | 14.8 | 16.0 | 10.3 | 33.1 | 16.3 | 70.4 | 23.3 | 78.2 | 31.0 | 80.3 |
| traffic light | 23.7 | 24.5 | 19.5 | 22.5 | 6.4 | 13.1 | 19.3 | 22.3 | 23.9 | 27.2 |
| pole | 8.7 | 8.8 | 34.1 | 39.4 | 19.2 | 21.5 | 33.8 | 38.9 | 35.4 | 41.0 |
| obs-str-bar-fallback | 0.9 | 0.9 | 0.9 | 0.9 | 0.0 | 0.0 | 0.0 | 0.0 | 6.4 | 9.5 |
| building | 45.3 | 60.0 | 51.8 | 87.1 | 26.1 | 49.0 | 52.0 | 83.9 | 54.3 | 82.6 |
| bridge | 43.3 | 44.2 | 43.3 | 44.2 | 31.6 | 60.4 | 31.6 | 60.4 | 8.1 | 44.7 |
| vegetation | 12.4 | 12.4 | 84.5 | 94.5 | 21.5 | 22.6 | 84.3 | 94.0 | 80.6 | 87.0 |
| sky | 91.4 | 92.8 | 94.6 | 98.8 | 84.1 | 93.8 | 94.7 | 98.7 | 82.3 | 85.3 |
| **Average** | 30.8 | 36.0 | 39.4 | 51.7 | 20.9 | 31.6 | 40.2 | 55.9 | 41.7 | 57.7 |

Table 11: Taxonomy mapping from GTA to Mapillary

| Source Label | Target Label Set |
|---|---|
| road | road, sidewalk, snow, sand, water, catch basin, manhole, rail track, lane marking-crosswalk, lane marking-general |
| sidewalk | sidewalk, curb, snow, sand, water |
| building | building, bridge, tunnel, phone booth, billboard, |
| wall | wall, bridge, tunnel, trash can, banner, billboard |
| fence | fence, guard rail |
| pole | pole, utility pole, trash can, banner, street light, traffic sign frame |
| traffic light | traffic light, street light |
| traffic sign | traffic sign (front), traffic sign (back), billboard, banner |
| vegetation | vegetation, snow |
| terrain | terrain, mountain, snow, sand, water |
| sky | sky |
| person | person |
| rider | bicyclist, motorcyclist |
| car | car, trailer, boat |
| truck | truck, caravan |
| bus | bus |
| train | on rails |
| motorcycle | motorcycle |
| bicycle | bicycle |
| unlabeled | bench, billboard, bridge, tunnel |

*Generate new class names within the context of street scene semantic segmentation, using the original class name as the head noun. Use synonyms or subcategories of the original class that make sense within this context, and if the class has multiple meanings, add specific context to avoid ambiguity. Please provide the original class names along with context names.*

Table 12: Taxonomy mapping from GTA to IDD

| Source Label | Target Label Set |
|---|---|
| road | road, sidewalk, drivable fallback |
| sidewalk | sidewalk, curb, drivable fallback, non-drivable fallback |
| building | building, bridge, billboard, |
| wall | wall, obs-str-bar-fallback, bridge, billboard |
| fence | fence, guard rail, obs-str-bar-fallback |
| pole | pole |
| traffic light | traffic light |
| traffic sign | traffic sign, billboard, banner |
| vegetation | vegetation, obs-str-bar-fallback |
| terrain | terrain, non-drivable fallback, obs-str-bar-fallback |
| sky | sky |
| person | person |
| rider | bicyclist, motorcyclist |
| car | car, autorickshaw |
| truck | truck, caravan |
| bus | bus |
| train | other vehicles |
| motorcycle | motorcycle |
| bicycle | bicycle |
| unlabeled | billboard, bridge |

For each label, we generate 10 context names. For labels without ambiguity, e.g. sky, we only use the original label for the text feature extraction.

Table 13: Context names for IDD labels

| Target Label | Context Names |
|---|---|
| road | road, main road, driving lane, paved road, highway, residential street, arterial road, rural road, city road, thoroughfare |
| drivable fallback | drivable terrain, traffic lane, vehicle lane, driveable path, car lane, driveable street, urban roadway, paved path, driveable surface, roadway |
| sidewalk | sidewalk, pavement, footpath, walkway, pedestrian path, side path, sidewalk pavement, urban sidewalk, street sidewalk, sidewalk lane, sidewalk area |
| non-drivable fallback | non-drivable terrain, pedestrian area, park path, garden path, bike lane, footpath, public plaza, grass area, green space, pedestrian walkway, non-driveable zone |
| person | person |
| rider | rider |
| motorcycle | motorcycle |
| bicycle | bicycle |
| autorickshaw | autorickshaw, three-wheeler, tuk-tuk, auto-rickshaw, motorized rickshaw, auto taxi, rickshaw, three-wheeled taxi, auto, motor tricycle, auto rickshaw |
| car | car, sedan, hatchback, coupe, convertible, SUV, sports car, station wagon, compact car, electric car, luxury car |
| truck | truck, pickup truck, semi-truck, delivery truck, dump truck, fire truck, tow truck, box truck, flatbed truck, garbage truck, tanker truck |
| bus | bus |
| vehicle fallback | other vehicles, train, tram, metro, trolleybus, light rail, cable car |
| curb | curb, road curb, sidewalk curb, curbside, street curb, pavement curb, curb edge, curb line, curb boundary, urban curb, curb strip |
| wall | wall, barrier wall, protective wall, retaining wall, boundary wall, perimeter wall, dividing wall, sound barrier wall, security wall, freestanding wall, partition wall |
| fence | fence, building fence, road fence, vehicle separation fence, pedestrian fence, safety fence, boundary fence, traffic fence, divider fence, protective fence, barrier fence |
| guard rail | guard rail, road guard rail, highway guard rail, safety guard rail, traffic guard rail, barrier guard rail, roadside guard rail, protective guard rail, metal guard rail, crash barrier, median guard rail |
| billboard | billboard, advertising billboard, roadside billboard, digital billboard, outdoor billboard, highway billboard, commercial billboard, urban billboard, street billboard, electronic billboard, large billboard |
| traffic sign | traffic sign, road sign, highway sign, street sign, regulatory sign, warning sign, directional sign, informational sign, traffic control sign, signpost, traffic marker |
| traffic light | traffic light, traffic signal, stoplight, traffic control light, intersection signal, traffic lamp, signal light, road signal, street light, traffic signal light, traffic control signal |
| pole | pole, street pole, lamp pole, traffic pole, sign pole, light pole, support pole, signal pole, flag pole, decorative pole, banner pole |
| obs-str-bar-fallback | obstructive structures and barriers, construction barrier, roadblock, traffic cone, temporary fence, safety barrier, barricade, obstruction, traffic barricade, road barrier, construction zone marker |
| building | building, structure, edifice, construction, residential building, commercial building, office building, apartment building, skyscraper, public building, urban building |
| bridge | road bridge, footbridge, pedestrian bridge, walking bridge, footpath bridge, foot crossing, small bridge, pedestrian crossing, walkway bridge, urban footbridge, trail bridge |
| vegetation | vegetation, urban vegetation, city greenery, roadside plants, street vegetation, urban foliage, city flora, park vegetation, public greenery, urban plants, green space |
| sky | sky |

Table 14: Context names for Mapillary labels (Part 1)

| Target Label | Context Names |
|---|---|
| Road | road, main road, driving lane, paved road, highway, residential street, arterial road, rural road, city road, thoroughfare |
| Snow | snow, snow pile, street snow, roadside snow, accumulated snow, snowbank, plowed snow, urban snow, compacted snow, snow drift, snow on pavement |
| Sand | sand, sand pile, street sand, roadside sand, piled sand, sandbank, accumulated sand, urban sand, sand on pavement, construction sand, loose sand |
| Catch Basin | catch basin, road catch basin, street catch basin, roadside catch basin, storm drain, drainage basin, sewer catch basin, street drain, gutter catch basin, road drain, stormwater basin |
| Manhole | manhole, road manhole, street manhole, sewer manhole, manhole cover, utility manhole, drainage manhole, storm drain manhole, roadside manhole, underground access, inspection manhole |
| Pothole | pothole, road pothole, street pothole, asphalt pothole, pavement pothole, highway pothole, surface pothole, pothole damage, roadway pothole, pothole crater, pothole on pavement |
| Bike Lane | bike lane, marked bike lane, roadside bike lane, main road bike lane, dedicated bike lane, paved bike lane, urban bike lane, bike path, protected bike lane, street bike lane, lane-marked bike lane |
| Rail Track | rail track, tram rail track, train rail track, street rail track, road rail track, urban rail track, tramway track, railroad track, commuter rail track, embedded rail track, rail track on pavement |
| Lane Marking - Crosswalk | crosswalk lane marking, street crosswalk marking, pedestrian crosswalk marking, zebra crossing marking, road crosswalk marking, intersection crosswalk marking, painted crosswalk, crosswalk lines, crosswalk road marking, sidewalk crosswalk marking |
| Lane Marking - General | general lane marking, road lane marking, street lane marking, highway lane marking, pavement lane marking, lane divider marking, traffic lane marking, lane line marking, roadway lane marking, lane boundary marking, asphalt lane marking |
| Water | water, urban water, river water, lake water, city river, roadside pond, street water, urban pond, city lake, small urban river, stormwater |
| Sidewalk | sidewalk, pavement, footpath, walkway, pedestrian path, side path, sidewalk pavement, urban sidewalk, street sidewalk, sidewalk lane, sidewalk area |
| Curb | curb, road curb, sidewalk curb, curbside, street curb, pavement curb, curb edge, curb line, curb boundary, urban curb, curb strip |
| Pedestrian Area | pedestrian area, street pedestrian area, pedestrian zone, pedestrian walkway, pedestrian street, urban pedestrian area, pedestrian plaza, pedestrian path, sidewalk pedestrian area, pedestrian crossing area, designated pedestrian area |
| Building | building, structure, edifice, construction, residential building, commercial building, office building, apartment building, skyscraper, public building, urban building |
| Bridge | road bridge, footbridge, pedestrian bridge, walking bridge, footpath bridge, foot crossing, small bridge, pedestrian crossing, walkway bridge, urban footbridge, trail bridge |
| Billboard | billboard, advertising billboard, roadside billboard, digital billboard, outdoor billboard, highway billboard, commercial billboard, urban billboard, street billboard, electronic billboard, large billboard |
| Tunnel | tunnel, road tunnel, tunnel entrance, highway tunnel, urban tunnel, vehicle tunnel, tunnel passage, tunnel opening, subway tunnel, underground tunnel, traffic tunnel |
| Wall | wall, barrier wall, protective wall, retaining wall, boundary wall, perimeter wall, dividing wall, sound barrier wall, security wall, freestanding wall, partition wall |
| Traffic Sign Frame | traffic sign frame, signpost frame, traffic sign holder, sign frame, sign support frame, road sign frame, traffic sign structure, sign mounting frame, sign frame support, traffic sign bracket |
| Trash Can | trash can, street trash can, public trash can, roadside trash can, outdoor trash can, urban trash can, sidewalk trash can, street garbage can, public waste bin, street litter bin, municipal trash can |
| Banner | banner, advertising banner, promotional banner, street banner, event banner, hanging banner, outdoor banner, banner sign, vertical banner, display banner, publicity banner |
| Fence | fence, building fence, road fence, vehicle separation fence, pedestrian fence, safety fence, boundary fence, traffic fence, divider fence, protective fence, barrier fence |
| Guard Rail | guard rail, road guard rail, highway guard rail, safety guard rail, traffic guard rail, barrier guard rail, roadside guard rail, protective guard rail, metal guard rail, crash barrier, median guard rail |
| Pole | pole, street pole, lamp pole, traffic pole, sign pole, light pole, support pole, signal pole, flag pole, decorative pole, banner pole |
| Utility Pole | utility pole, electric pole, telephone pole, power pole, transmission pole, cable pole, utility line pole, utility post, service pole, communication pole, distribution pole |
| Street Light | street light, street lamp, road light, streetlight, lamp post, street lighting, urban street light, sidewalk light, public street light, street lantern, street illumination |
| Front Side Of Traffic Sign | front side of traffic sign, traffic sign front, front face of traffic sign, sign front, traffic sign face, front panel of traffic sign, signboard front, traffic sign display, front view of traffic sign, sign front side, traffic sign surface |
| Back Side Of Traffic Sign | back side of traffic sign, traffic sign back, back face of traffic sign, sign back, rear of traffic sign, signboard back, traffic sign reverse, sign back panel, back side of sign, traffic sign rear view, reverse side of traffic sign |
| Traffic Light | traffic light, traffic signal, stoplight, traffic control light, intersection signal, traffic lamp, signal light, road signal, street light, traffic signal light, traffic control signal |
| Vegetation | vegetation, urban vegetation, city greenery, street vegetation, roadside vegetation, urban plants, city foliage, urban flora, street greenery, public vegetation, cityscape vegetation |

Table 15: Context names for Mapillary labels (Part 2)

| Target Label | Context Names |
|---|---|
| Terrain | terrain, urban terrain, city landscape, street terrain, roadside terrain, urban ground, cityscape terrain, urban land, urban surface, city terrain, urban topography |
| Mountain | mountain, mountain peak, mountain range, mountain slope, rocky mountain, highland mountain, mountain summit, alpine mountain, mountain ridge, forest mountain, mountain terrain |
| Sky | sky |
| Person | person |
| Bicyclist | bicyclist, bike rider, cyclist, bicycle rider, bicycle commuter, mountain biker, road cyclist |
| Motorcyclist | motorcyclist, motorcycle rider, motorcycle driver, motorbike rider, motorcycle commuter, road motorcyclist |
| Car | car, sedan, hatchback, coupe, convertible, SUV, sports car, station wagon, compact car, electric car, luxury car |
| Trailer | trailer, utility trailer, travel trailer, cargo trailer, flatbed trailer, camper trailer, enclosed trailer, livestock trailer, dump trailer |
| Boat | boat, sailboat, motorboat, fishing boat, speedboat, yacht, canoe, kayak, pontoon boat, dinghy, houseboat |
| Truck | truck, pickup truck, semi-truck, delivery truck, dump truck, fire truck, tow truck, box truck, flatbed truck, garbage truck, tanker truck |
| Caravan | caravan, travel caravan, camper caravan, motorhome, touring caravan, RV (recreational vehicle), fifth-wheel caravan, pop-up caravan, teardrop caravan, static caravan, off-road caravan |
| Bus | bus |
| On Rails | on rails |
| Motorcycle | motorcycle |
| Bicycle | bicycle |
| Bench | bench, street bench, public bench, park bench, sidewalk bench, outdoor bench, urban bench, pavement bench, city bench, public seating bench, roadside bench |

