# OpenReview forum: "DynAlign: Unsupervised Dynamic Taxonomy Alignment for Cross-Domain Segmentation"
_ICLR.cc/2025/Conference — ICLR 2025 Poster_

### Official Review · Reviewer_7s4Z · 2024-11-02

**Soundness:** 3
**Presentation:** 2
**Contribution:** 2
**Rating:** 6
**Confidence:** 2

**Summary:**

Traditional UDA methods for semantic segmentation often assume identical class labels in both source and target domains. To tackle this, this paper presents DynAlign, a two-stage framework that combines UDA with foundation models to address both image-level and label-level domain gaps. It leverages prior semantic knowledge to align source and target categories and uses foundation models for detailed segmentation and category realignment. To boost accuracy further, a knowledge fusion strategy is divised to dynamically adapt to diverse scene contexts.

**Strengths:**

1. The method design of this paper is straightforward and easy to understand, with sufficient details and reliable techniques.
2. The paper aims to address the label space mismatch issue in UDA tasks, and this research problem is very meaningful.
3. The research motivation of the paper is strong, and the solution is clearly articulated.

**Weaknesses:**

1. The paper dedicates considerable space to comparisons with open-vocabulary semantic segmentation methods and benchmarks its approach against those used in that task. I understand that this is due to the limitations in label space, which restrict traditional UDA methods. However, since the focus of this paper is UDA, I believe the authors should concentrate on comparisons with UDA methods to better highlight the advantages of this work. Throughout my reading, I found myself constantly switching between the two tasks (UDA and open-vocabulary SS), making it challenging to clearly grasp the contributions of this paper.

2. I understand that earlier methods also utilized foundation models but still required target labels for few-shot training. In contrast, this paper employs foundation models in a fully unsupervised manner, correct? Is this the main difference/contribution this paper has compared to previous methods?

3. How to better utilize foundation models has been a significant research topic in recent years, and I find the authors' exploration of this subject very meaningful. However, with the increasing number of foundation models available, it is insufficient to merely leverage more or better foundation models to complete tasks. The authors should emphasize the specific benefits of their design rather than simply implementing it in an unsupervised manner. Since the inference of large models is inherently unsupervised, simply using them does not constitute a substantial contribution.

4. The use of foundation models still incurs computational resource demands, which can affect the practical applicability of this framework. The experimental section should include comparisons regarding the computational resource consumption of the models, such as memory usage and inference time.

**Questions:**

See weaknesses.

---

> ### Author Response · Authors · 2024-11-22
>
> ***Q1: Confusion between UDA and open-vocabulary methods***
>
> Thank you for your thoughtful feedback. The contribution of this paper lies exactly in addressing the intersection of UDA and open-vocabulary setups within a single setup. In this work, we try to solve the **unsupervised taxonomy adaptive domain adaptation problem** with challenges in two key aspects: 1) **Unsupervised Domain Adaptation (UDA):** While UDA methods address domain shifts, they typically assume consistent label spaces, making them inflexible for handling label-space discrepancies. 2) **Taxonomy Discrepancies:** Adapting to mismatched label spaces—arising from differences in granularity, definitions, or new categories—remains challenging. Recent open-vocabulary approaches built on foundation models prioritize open-world knowledge but often overlook the integration of domain-specific knowledge critical for specialized tasks.
> To conclude, most existing research and benchmarks have tackled these challenges separately (e.g., standard UDA assumes consistent label spaces, while taxonomy adaptation often overlooks domain shifts). Consequently, the intersection of these two problems remains a relatively new area with limited prior work. Our study highlights this gap and aims to provide a flexible framework to address both challenges simultaneously for more robust solutions to this real-world problem.
>
> To address the concern about comparisons with UDA methods, we compare our performance with the traditional UDA setting below. As explained in Lines 410-411, _Traditional DA methods are limited to fixed label spaces, making them inapplicable to our problem setting_. Therefore, we compare the results for known classes of HRDA under the traditional UDA setting, adapted from GTA to the Mapillary dataset. The results show that our framework effectively preserves domain knowledge for the known classes while enriching prior knowledge for the unknown classes.
>
> |settings|All classes| |Known classes| |Unknown classes| |
> |--------|-----------|-|-------------|-|---------------|-|
> | Setting             | mIoU   | mAcc   | mIoU   | mAcc   | mIoU   | mAcc   |
> |GTA to Mapillary (HRDA)|N/A|N/A|65.8|75.8|N/A|N/A|
> |GTA to Mapillary (ours)|36.7|53.0|62.4|72.6|19.6|39.9|
>
> These results demonstrate that our approach is competitive even under traditional UDA settings while retaining the flexibility to handle open-vocabulary setups. This highlights the advantage of our framework in bridging these two tasks effectively.
>
> ***Q2: I understand that earlier methods also utilized foundation models but still required target labels for few-shot training. In contrast, this paper employs foundation models in a fully unsupervised manner, correct? Is this the main difference/contribution this paper has compared to previous methods?***
>
> Thank you for raising this question. We would like to clarify that 1) the previous taxonomy-adaptive domain adaptation methods do not use foundation models, but few-shot in-domain labeled images for training. For example, TACS [1] uses few-shot images for each novel class to learn the new label distribution. Instead, our method leverages foundation models to incorporate its prior knowledge to replace the in-domain few-shot samples in a fully unsupervised method. 2) recent open-vocabulary methods leverage foundation models' prior knowledge for segmentation, but do not incorporate any in-domain knowledge. This leads to their inferior performance when it comes to specific domains.
>
> To conclude, our main contribution does not solely lying in its unsupervised manner, but also in proposing a new benchmark and approach to leverage the prior knowledge of foundation models to stengthen the in-domain prediction and make it scalable to unknown classes.

---

> ### Author Response · Authors · 2024-11-22
>
> ***Q3: How to better utilize foundation models has been a significant research topic in recent years, and I find the authors' exploration of this subject very meaningful. However, with the increasing number of foundation models available, it is insufficient to merely leverage more or better foundation models to complete tasks. The authors should emphasize the specific benefits of their design rather than simply implementing it in an unsupervised manner. Since the inference of large models is inherently unsupervised, simply using them does not constitute a substantial contribution.***
>
> We thank you for your thoughtful comments. We agree that simply leveraging foundation models does not inherently lead to superior performance or constitute a significant contribution in specific tasks. The primary contribution of our work lies in effectively fusing domain-specific knowledge with the prior knowledge of foundation models.
>
> As highlighted in Table 1, domain-specific knowledge remains crucial for achieving robust performance. For example, our results compared to Grounded-SAM [2]—which relies solely on foundation models for open-vocabulary semantic segmentation—demonstrate that incorporating domain-specific knowledge through UDA significantly enhances performance, even when using strong foundation models for open-vocabulary tasks.
>
> While foundation models are inherently unsupervised and designed for general-purpose tasks, the novelty of our approach lies in adapting these models to domain-specific challenges via a modular framework. To validate this, we tested the performance by substituting different modules within our framework and reported the results in the table below. By testing multiple configurations (e.g., HRDA → MIC, SAM → MobileSAM), we show that our framework does not rely on simply using "better" foundation models. Instead, it provides an effective and flexible way to integrate any general foundation models into domain-specific tasks, making the framework in a practical and adaptable way.
>
> To summarize, our work is the first to implement and benchmark unsupervised taxonomy-adaptive domain adaptation, addressing the underexplored intersection of open-vocabulary tasks and domain adaptation. We hope that releasing our code and benchmark setup will serve as a useful resource for the community. Additionally, 'better' foundation models could improve performance in open-vocabulary setups, our results clearly demonstrate that combining these models with domain-specific knowledge significantly improves performance in road-scene segmentation.
>
> | Method             | mIoU   | mAcc   |
> |---------------------|--------|--------|
> | HRDA -> MIC[3]      | 37.7 |  54.4   |
> | SAM -> MobileSAM[4]       | 35.2  | 51.5  |
> | CLIP -> SCLIP[5]      |   36.0     |   50.9     |
> |GPT-4 -> Llama[6] |37.5 |52.1|
> |Ours (in paper)|36.7|53.0|

---

> ### Author Response · Authors · 2024-11-22
>
> ***Q4: The experimental section should include comparisons regarding the computational resource consumption of the models, such as memory usage and inference time.***
>
> Thank you for raising this question. Our method comprises three main components: the UDA semantic segmentation model (HRDA), SAM, and CLIP. For semantic segmentation on high-resolution images, the memory usage primarily depends on the semantic segmentation model (HRDA), while the inference time is influenced by the reassignment of novel class labels. We provide detailed information on memory usage, inference time, and model parameters in the table below.
>
> The total model parameter count of our method includes the UDA model parameters, plus 308M for SAM (ViT-L) and 351M for CLIP (ConvNeXt-Large). The reported total memory usage corresponds to the memory allocated during the complete inference process for a single image. We anticipate that further memory optimizations could be achieved by refining the code base, particularly by cleaning up intermediate tensor caches.
>
> Overall, our framework requires reasonable computational resources. Moreover, the UDA framework can be flexibly substituted to accommodate memory limitations. As shown in the table, replacing HRDA with DAFormer significantly reduces memory consumption. For efficient inference, a segmentation model can also be trained using pseudo-labels generated by our framework, as described in *Section 5.3*. For example, a trained Mask2Former model using our inference pseudo labels, indicated as _Ours (pseudo-label, Mask2Former)_, achieves inference in just **0.3 seconds** per single image, offering a significant efficiency boost while maintaining high performance.
>
>
> |model|memory usage|inference time|model parameter|
> |-----|------|------|------|
> |HRDA| 23.9 GB | 1.9 s | 86 M |
> |DynAlign (HRDA)|34.8 GB | 64.1 s | 668 M|
> |DAFormer| 10.5 GB | 0.7 s | 86 M |
> |DynAlign (DAFomer)|21.7 GB | 64.1 s | 668 M|
> |Ours (pseudo-label, HRDA)| 23.9 GB | 1.9 s | 86 M |
> |Ours (pseudo-label, DAFormer)| 10.5 GB | 0.7 s | 86 M |
> |Ours (pseudo-label, Mask2Former)|3.7 GB | 0.3 s | 215 M|
> >NOTE: For the inference time, the performance on average of 10 runs on a single NVIDIA A100-SXM4-80GB is reported.
>
> [1] Gong, Rui, et al. Tacs: Taxonomy adaptive cross-domain semantic segmentation. ECCV 2022.
>
> [2] Ren, Tianhe, et al. Grounded sam: Assembling open-world models for diverse visual tasks. arXiv preprint arXiv:2401.14159 (2024).
>
> [3] Hoyer, Lukas, et al. MIC: Masked image consistency for context-enhanced domain adaptation. CVPR 2023.
>
> [4] Zhang, Chaoning, et al. Faster segment anything: Towards lightweight sam for mobile applications. arXiv preprint arXiv:2306.14289 (2023).
>
> [5] Wang, Feng, Jieru Mei, and Alan Yuille. Sclip: Rethinking self-attention for dense vision-language inference. ECCV 2025.
>
> [6] Touvron, Hugo, et al. Llama: Open and efficient foundation language models. arXiv preprint arXiv:2302.13971 (2023).

---

> ### Comment · Reviewer_7s4Z · 2024-11-25
>
> I appreciate the authors' efforts in their rebuttal, which addressed most of my concerns, leading me to raise my score. While the rebuttal provides strong insights, these are not sufficiently reflected in the current version of the paper. As such, I am not fully confident in my revised score. If the paper is accepted, significant revisions will be necessary to better integrate the insights presented in the rebuttal.

---

> > ### Author Response · Authors · 2024-11-26
> >
> > Thank you for your reply and for raising the score! We sincerely apologize for the delay in modifying our paper, as we misunderstood certain aspects of the rebuttal format. To address this, we have uploaded another revised version that includes the following improvements:
> >
> > 1. **Motivation and Contribution**: We have added further explanations on the novelty and contributions of our paper, which can be found in _Lines 92–97, Lines 153–155, and Lines 440-444_. We hope these additions further clarify and strengthen the motivation and novelty of our work. We also include the comparison of UDA method in **Table 1**.
> >
> > 2. **Distinction from Existing Methods**: We emphasize the difference between our method and existing taxonomy-adaptive methods in _Lines 153-155_, highlighting that these approaches do not incorporate foundation models.
> >
> > 3. **Difference from Foundation-Model-Only Methods**: We have strengthened the distinction between our approach and methods that purely rely on foundation models in _Lines 92-97_. In the experimental section, we have added more analysis in _Lines 440-444_ to underscore the advantages of our method compared to foundation-model-based approaches. Additionally, we include the experiment on module ablations in _Lines 495-501_ and **Table 4** to demonstrate our method’s effectiveness in flexibly fusing domain-specific knowledge with the prior knowledge of foundation models.
> >
> > 4. **Memory Usage and Efficiency**: The experiment on memory usage and efficiency has been highlighted in ***Section 5.3***, with detailed information included in the appendix ***A.2 COMPUTATIONAL EFFICIENCY*** (_Lines 854–882_).
> >
> > We deeply appreciate the discussions you initiated, which have significantly helped us refine our contributions and better position our work at the intersection of traditional UDA and open-vocabulary approaches. These insights have been thoughtfully incorporated into the latest revision.
> >
> > Please let us know if there are specific parts of our discussion that you feel could be further emphasized to enhance the clarity and readability of our work. We are more than happy to engage in further discussions or address any related questions.
> >
> > Thank you once again for your invaluable feedback and for helping improve the quality of our work!

---

### Official Review · Reviewer_GEaE · 2024-11-04

**Soundness:** 3
**Presentation:** 3
**Contribution:** 2
**Rating:** 6
**Confidence:** 4

**Summary:**

This paper presents  a two-stage framework that integrates UDA with foundation models to bridge both the image-level and label-level domain gaps. It consists of multiple modules including conventional UDA to bridge the distribution gap between source and target, vision foundation model to extract general segmentation, LLM to construct the label taxonomy, and CLIP to align visual and text features to generate well-aligned pseudo labels for self-training.

**Strengths:**

Belows are strong points that this paper has:

1. Well-written and systematically organized to make the reader to understand more easily.

2. It proposes reasonable fusion strategy between UDA and Foundation model to resolve the distribution and label shift issue  in cross-domain scenario

3. It achieves remarkable performance improvement, showing the effectiveness of the proposed method on two benchmark.

4. Figure 4 shows that training with generated pseudo labels can not only achieve better score but also increase the inference efficiency.

**Weaknesses:**

Belows are weak points that this paper has:

1. In order to align with line 112 ("integrating with any UDA-based semantic segmentation models"), it would be better that author provide another examples to show the compatibility of the proposed method (e.g., MIC + DynAlign, or other UDA model + DynAlign)

2. Need to have more analysis on how robust the proposed knowledge fusion is. One way to do is that using different foundation model (e.g., MobileSAM for SAM, Llama for GPT-4,  another VLM for CLIP) can also work with the proposed method. Showing this compatibility can provide more flexible fusion environment.

3. Any failure case while fusing UDA and foundation model?

**Questions:**

Please see above Weaknesses.

---

> ### Author Response · Authors · 2024-11-22
>
> ***Q1-2: Integration with other methods: it would be better that author provide another examples to show the compatibility of the proposed method (e.g., MIC + DynAlign, or other UDA model + DynAlign) and use different foundation model (e.g., MobileSAM for SAM, Llama for GPT-4, another VLM for CLIP) can also work with the proposed method to show this compatibility can provide more flexible fusion environment.***
>
> Thank you for suggesting this meaningful experiment! In the table below, we present results on the Mapillary dataset, where we substitute individual modules in our framework (e.g., replacing HRDA with MIC). The results demonstrate the flexibility of our method to work effectively in combination with various existing models. Due to time constraints, we implemented only one substitution per module for this study. However, we acknowledge the value of a more comprehensive evaluation and will include additional substitution experiments in the final version of the paper.
>
>
> | Method             | mIoU   | mAcc   |
> |---------------------|--------|--------|
> | HRDA -> MIC[1]      | 37.7 |  54.4   |
> | SAM -> MobileSAM[2]       | 35.2  | 51.5  |
> | CLIP -> SCLIP[3]      |   36.0     |   50.9     |
> |GPT-4 -> Llama[4] |37.5 |52.1|
> |Ours (in paper)|36.7|53.0|
>
> ***Q3: Any failure case while fusing UDA and foundation model?***
>
> Due to the flexibility of our method, any UDA model or foundation model can be substituted into the corresponding module, as demonstrated in the response above. Potential failure cases may arise when foundation models, such as SAM or CLIP, are applied to domains significantly different from those encountered during their pre-training, such as medical image segmentation. This challenge could potentially be addressed by using foundation models specifically developed for specialized fields [5, 6]. Nevertheless, their performance in this context needs to be thoroughly tested and validated.
>
> [1] Hoyer, Lukas, et al. MIC: Masked image consistency for context-enhanced domain adaptation. CVPR 2023.
>
> [2] Zhang, Chaoning, et al. Faster segment anything: Towards lightweight sam for mobile applications. arXiv preprint arXiv:2306.14289 (2023).
>
> [3] Wang, Feng, Jieru Mei, and Alan Yuille. Sclip: Rethinking self-attention for dense vision-language inference. ECCV 2025.
>
> [4] Touvron, Hugo, et al. Llama: Open and efficient foundation language models. arXiv preprint arXiv:2302.13971 (2023).
>
> [5] Zhang, Yunkun, et al. Text-guided foundation model adaptation for pathological image classification. MICCAI 2023.
>
> [6] Chen, Zeming, et al. Meditron-70b: Scaling medical pretraining for large language models. arXiv preprint arXiv:2311.16079 (2023).

---

> > ### Comment · Reviewer_GEaE · 2024-11-28
> >
> > Thank you to the authors for providing additional insights and experiments. They have addressed my concerns, and I will maintain my recommendation for acceptance. I recommend the authors integrate this additional insight into final manuscript.

---

### Official Review · Reviewer_cqS3 · 2024-11-04

**Soundness:** 2
**Presentation:** 4
**Contribution:** 2
**Rating:** 6
**Confidence:** 5

**Summary:**

The paper targets the label-level domain gap, which is common in real-world scenarios.  The authors introduce DynAlign, a
two-stage framework that integrates UDA with foundation models to bridge both the image-level and label-level domain gaps. DynAlign leverages prior semantic knowledge to align source categories with target categories that can be novel, more fine-grained, or named differently (e.g., ‘vehicle’ to {‘car’, ‘truck’, ‘bus’}). Foundation models are then employed for precise segmentation and category reassignment. To further enhance accuracy, the authors propose a knowledge fusion approach that dynamically adapts to varying scene contexts. DynAlign generates accurate predictions in a new target label space without requiring any manual annotations.

**Strengths:**

1. The problem in this paper is of great practical significance.

2. The writing of this paper is clear and well-organized.

3. The experimental results showed some improvements when using such technology, compared with previous rivals.

**Weaknesses:**

1. The article only explores coarse-to-fine experiments. In real-world scenarios, however, the label structure of the target domain may sometimes be coarser than that of the source domain. How would the proposed method be adapted for such a setting? Additionally, the effectiveness of the method under these conditions has not been verified.

2. The contribution towards DG area is limited. The generalization achieved by the methods in this paper primarily derives from the pre-trained models—specifically, SAM's generalization in segmentation and CLIP's generalization in multimodal alignment—rather than from the techniques introduced within the paper itself. As a result, this work mainly resembles a mapping technique that aligns open-vocabulary labels between the source and target domains, an idea that has been explored extensively in other studies. Consequently, I do not view this paper as making a significant contribution to the field of domain generalization. In my opinion, as an ICLR article, it still needs a certain theoretical depth.

Minor issues:
Some papers are cited repeatedly in the reference, such as "Domain aligned clip for few-shot classification." and "Scaling open-vocabulary image segmentation with image-level labels."

**Questions:**

Please see the "Weaknesses" part.

---

> ### Author Response · Authors · 2024-11-22
>
> ***Q1: In real-world scenarios, the label structure of the target domain may sometimes be coarser than that of the source domain. How would the proposed method be adapted for such a setting?***
>
> We thank the reviewer for pointing this out. To address this concern, we have conducted additional experiments under **Fine-to-Coarse Setting:** Here, the target domain labels are coarser than the source domain labels. For this, we adapt from Mapillary (45 classes) to Cityscapes (19 classes). The results demonstrate that our framework produces robust predictions under the fine-to-coarse setting, effectively handling scenarios where the label spaces are not identical.
>
>
> | Setting             | mIoU   | mAcc   |
> |---------------------|--------|--------|
> | GTA to Cityscapes (HRDA, 19 → 19)       | 74.9  | 82.0  |
> | Mapillary to Cityscapes (ours, 45 ->19)      |   70.4    |   81.5     |
>
> ***Q2: The contribution towards DG area is limited.***
>
>
> We thank you for your for highlighting areas of concern. While our framework incorporates pre-trained models such as SAM and CLIP, the primary contribution lies in integrating their general knowledge with domain-specific knowledge from UDA. This fusion enables the framework to address real-world scenarios where the generalization capabilities of foundation models alone are insufficient, as demonstrated in *Table 1*. Therefore, our work emphasizes strengthening UDA in-domain performance through this integration, rather than leveraging or improving the generalization ability of pre-trained models themselves.
>
> We would also like to clarify that the taxonomy mapping serves as one step in our framework to establish the initial label-space correlation. Beyond this, we contribute an effective approach to combine domain knowledge and prior knowledge by fusing their predictions. Additionally, we have compared our method with others, such as Grounded-SAM [1], which maps open-vocabulary labels to SAM-segmented regions. The results validate the effectiveness of our approach in leveraging in-domain knowledge to improve the performance. We thank you again for opening this discussion and are willing to discuss further or conduct additional ablations to compare with more related works.
>
> We acknowledge that this paper places greater emphasis on practical contributions than on theoretical developments. Specifically, we introduce a new benchmark and experimental setup to evaluate taxonomy adaptive domain adaptation under both domain and label-space shifts. Our work is an initial attempt at addressing these challenges through a robust and practical approach, which we believe also contributes to the ICLR community. We are happy to receive suggestions or more discussions on possible theoretical analyses that could be interesting to explore in future in-depth follow-up studies and thank you again for your insightful feedback.
>
> [1] Ren, Tianhe, et al. Grounded sam: Assembling open-world models for diverse visual tasks. arXiv preprint arXiv:2401.14159 (2024).
>
> ***Minor issues***
> We thank the reviewer, and have fixed them in the updated main document.

---

> > ### Comment · Reviewer_cqS3 · 2024-11-27
> >
> > Thank you for your response. The additional Fine-to-Coarse experiments have proven that the proposed method can be applied to more realistic scenarios. Although the method lacks some theoretical support, the practical contributions are impressive. I would raise my score to 6.

---

> > ### Author Response · Authors · 2024-11-27
> >
> > Thank you for recognizing the significance and practical contributions of this work and for raising the score! We sincerely appreciate your insightful suggestions regarding the motivation and experiments, which have greatly strengthened our work.

---

### Official Review · Reviewer_ddgh · 2024-11-04

**Soundness:** 3
**Presentation:** 2
**Contribution:** 2
**Rating:** 6
**Confidence:** 4

**Summary:**

This paper addresses the label-space domain gap within cross-domain semantic segmentation by proposing a novel framework that integrates traditional unsupervised domain adaptation (UDA) methods with foundational models such as large language models, vision-language models, and vision foundation models. Specifically, the large language model is utilized to reason and establish mappings between source and target label spaces. Using these mappings, predictions from unsupervised adaptation segmentation models are transformed into the target label space and subsequently fused with predictions from foundation models, including SAM and CLIP. The proposed framework demonstrates superior performance compared to several open-vocabulary recognition models.

**Strengths:**

1. The proposed framework is technically sound.
2. The paper addresses a practical and significant issue: the label-space domain gap within domain adaptation.

**Weaknesses:**

1. High Complexity and Heuristic Dependence
The framework’s complexity, along with its reliance on various heuristics, may hinder practical usability. Additionally, some heuristics lack clear specifications, making reproduction challenging. Key points include:

a) Inference Complexity: The framework requires inference from at least three models (UDA model, SAM, and CLIP), contributing to increased computational demands and overall complexity.

b) Semantic Taxonomy Mapping: The framework initially prompts GPT-4 for potential taxonomy mappings, refined through human input. This process adds a human burden due to prompt tuning and taxonomy refinement and is only briefly outlined in the supplementary material (lines 916 & 947), lacking comprehensive detail.

c) Criteria for Label Ambiguity: The paper states, “For labels without ambiguity, e.g., sky, we only use the original label for text feature extraction” (lines 1000-1001). The criteria for “ambiguity” remain unclear, introducing additional heuristics or human intervention.

2. Limited Experimental Scope
While the paper showcases superior performance in certain settings, the experimental scope and choice of baselines are somewhat limited. Expanding the evaluation would strengthen the framework’s claim of superiority:

a) Additional Baselines with Foundation Models: Numerous studies have leveraged foundation models for semantic segmentation, yet many of these are omitted in the experimental section. Including models such as [A] for domain-generalized segmentation and [B] for open-vocabulary segmentation would enhance comparability.

b) Traditional Domain Adaptation Settings: Although the framework targets handling substantial label-space domain gaps for real-world applications, demonstrating its efficacy in traditional domain adaptation settings (e.g., GTA5 to Cityscapes, with a smaller label-space gap) is also important.

c) Other Label-Space Domain Gaps: The experiments primarily focus on scenarios where the target domain includes more semantic classes than the source (e.g., 45 classes for Mapillary, 24 for IDD, and 19 for GTA). Evaluating scenarios with fewer classes in the target domain (e.g., Mapillary to Cityscapes) would substantiate the framework’s generalizability.

References

[A] Stronger, Fewer, & Superior: Harnessing Vision Foundation Models for Domain Generalized Semantic Segmentation, CVPR 2024

[B] Open-Vocabulary SAM: Segment and Recognize Twenty-thousand Classes Interactively, ECCV 2024

**Questions:**

The framework seems to rely purely on SAM for pixel grouping, with UDA and CLIP then reclassifying each mask region. Is this my understanding accurate?

---

> ### Author Response · Authors · 2024-11-22
>
> ***Q1: Inference complexity***
>
> We thank you for raising this concern. We provide details on the memory usage, inference time, and model parameters in the table below for your reference. The total model parameter count of our method includes the UDA model parameters, plus 308M for SAM (ViT-L) and 351M for CLIP (ConvNeXt-Large). The reported total memory usage corresponds to the memory allocated during the complete inference process for a single image. We anticipate that further memory optimizations could be achieved by refining the code base, particularly by cleaning up intermediate tensor caches.
>
> Overall, our framework requires reasonable computational resources. Moreover, the UDA framework can be flexibly substituted to accommodate memory limitations. As shown in the table, replacing HRDA with DAFormer significantly reduces memory consumption. For efficient inference, a segmentation model can also be trained using pseudo-labels generated by our framework, as described in *Section 5.3*. For example, a trained Mask2Former model using our inference pseudo labels, indicated as _Ours (pseudo-label, Mask2Former)_, achieves inference in just **0.3 seconds** per single image, offering a significant efficiency boost while maintaining high performance.
>
> |model|memory usage|inference time|model parameter|
> |-----|------|------|------|
> |HRDA| 23.9 GB | 1.9 s | 86 M |
> |ours (HRDA)|34.8 GB | 64.1 s | 668 M|
> |DAFormer| 10.5 GB | 0.7 s | 86 M |
> |ours (DAFomer)|21.7 GB | 64.1 s | 668 M|
> |Ours (pseudo-label, HRDA)| 23.9 GB | 1.9 s | 86 M |
> |Ours (pseudo-label, DAFormer)| 10.5 GB | 0.7 s | 86 M |
> |Ours (pseudo-label, Mask2Former)|3.7 GB | 0.3 s | 215 M|
> >NOTE: For the inference time, the performance on average of 10 runs on a single NVIDIA A100-SXM4-80GB is reported.
>
> Regarding the combination of the UDA model, SAM, and CLIP, we would like to emphasize that our framework does not involve a complex integration strategy. Instead, it is modular, allowing straightforward combination and substitution of individual components. As demonstrated in the table below with results on the Mapillary dataset, we substituted specific modules (e.g., replacing HRDA with MIC), and the results highlight the flexibility of our method to integrate effectively with various existing models. Overall, while our framework comprises multiple components, it is not strategically complex. On the contrary, it is flexible and practical across diverse scenarios.
>
> | Method             | mIoU   | mAcc   |
> |---------------------|--------|--------|
> | HRDA -> MIC[1]      | 37.7 |  54.4   |
> | SAM -> MobileSAM[2]       | 35.2  | 51.5  |
> | CLIP -> SCLIP[3]      |   36.0     |   50.9     |
> |GPT-4 -> Llama[4] |37.5 |52.1|
> |Ours (in paper)|36.7|53.0|
>
> ***Q2: Semantic Taxonomy Mapping: The framework initially prompts GPT-4 for potential taxonomy mappings, refined through human input. This process adds a human burden due to prompt tuning and taxonomy refinement and is only briefly outlined in the supplementary material (lines 916 & 947), lacking comprehensive detail.***
>
> We thank the reviewer for highlighting this important issue. Our framework utilizes GPT-4 to generate an initial proposal for potentially correlated taxonomy mappings between the source and target domains. However, we acknowledge that human intervention is necessary to refine these mappings due to the differing definitions of classes across datasets. For example, in the Mapillary dataset, the term *"guard rail"* specifically refers to roadside guard rails, whereas in other datasets or in general usage, it may represent a broader *guarding fence*. These discrepancies underscore the importance of incorporating human understanding to ensure that the mappings are accurate and contextually appropriate.
>
> We recognize that this process adds a degree of human involvement and that the results of the framework can be influenced by the quality of these mappings. However, it is important to emphasize that the involvement of human expertise in this process is minimal. The taxonomy mapping only needs to be defined once per dataset pair and involves verifying and refining the automatically proposed mappings, which is a significantly smaller effort compared to tasks such as pixel-wise manual annotation. This small but essential step ensures consistency and reliability in the results without imposing a substantial burden on practitioners.

---

> ### Author Response · Authors · 2024-11-22
>
> ***Q3: Criteria for Label Ambiguity: The paper states, “For labels without ambiguity, e.g., sky, we only use the original label for text feature extraction” (lines 1000-1001). The criteria for “ambiguity” remain unclear, introducing additional heuristics or human intervention.***
>
> Thank you for raising this concern. For labels with inherent ambiguity, we refer to classes that are broad or highly context-dependent. For example, the class *"other vehicles"* may encompass various types of vehicles that are not explicitly defined in the dataset, making precise labeling a challenge. Similarly, the class *"ego vehicle"* refers specifically to the vehicle equipped with the camera, which often appears only partially in the image. This adds complexity, as the visual cues for identifying this class are limited and heavily reliant on context. Such ambiguous labels are difficult to describe accurately in language because their meaning often depends on the dataset’s specific context or visual representation. This highlights the importance of careful consideration during the mapping process to ensure accurate and meaningful interpretations. In datasets with more domain-specific expert knowledge, these ambiguous classes can be better defined and mapped appropriately.
>
>
> ***Q4: Additional Baselines with Foundation Models: Numerous studies have leveraged foundation models for semantic segmentation, yet many of these are omitted in the experimental section. Including models such as [A] for domain-generalized segmentation and [B] for open-vocabulary segmentation would enhance comparability.***
>
> We thank you for mentioning the important works for comparison. For work [A] in domain generalized segmentation, as it does not address label-level adaptation, a direct comparison is not feasible. This work [A] leverages the foundational models to mitigate the domain gap on the image level.  We thank you for highlighting this correlated work and have updated our paper to include a discussion on this work in the related works section. For work [B] on open-vocabulary segmentation, we are in the process of adapting it to the street scene dataset used in our study and will update the experimental results later.
>
> ***Q5: Traditional Domain Adaptation Settings: Although the framework targets handling substantial label-space domain gaps for real-world applications, demonstrating its efficacy in traditional domain adaptation settings (e.g., GTA5 to Cityscapes, with a smaller label-space gap) is also important. and Other Label-Space Domain Gaps: The experiments primarily focus on scenarios where the target domain includes more semantic classes than the source (e.g., 45 classes for Mapillary, 24 for IDD, and 19 for GTA). Evaluating scenarios with fewer classes in the target domain (e.g., Mapillary to Cityscapes) would substantiate the framework’s generalizability.***
>
>
> We thank the reviewer for pointing this out. In the first table below, we present results for the following settings: 1) **Traditional UDA Setting:** In this case, the target label space is identical to the source label space. We adapt from GTA to Cityscapes, both using the same label space. 2) **Fine-to-Coarse Setting:** Here, the target domain labels are coarser than the source domain labels. For this, we adapt from Mapillary (45 classes) to Cityscapes (19 classes). In both the traditional UDA and fine-to-coarse settings, where sufficient domain knowledge about the label space is available, our framework preserves the in-domain performance and provides slight improvements when adequate in-domain knowledge is already present.
>
> In the second table, we present results under **taxonomy-adaptive domain adaptation** with a smaller label-space domain gap. Specifically, we adapt from Synthia (16 classes) to Cityscapes (19 classes). These results highlight that our method not only maintains strong performance on known classes but also predicts effectively for unknown classes.
>
> | Setting             | mIoU   | mAcc   |
> |---------------------|--------|--------|
> | GTA to Cityscapes (HRDA, 19 → 19)       | 74.9  | 82.0  |
> | GTA to Cityscapes (Ours, 19 → 19)       | 75.9  | 83.7  |
> | Mapillary to Cityscapes (ours, 45 ->19)      |   70.4    |   81.5     |
>
> |settings|All classes| |Known classes| |Unknown classes| |
> |--------|-----------|-|-------------|-|---------------|-|
> | Setting             | mIoU   | mAcc   | mIoU   | mAcc   | mIoU   | mAcc   |
> |Synthia to Cityscapes (HRDA, 16 -> 16)|N/A|N/A|66.8|73.7|N/A|N/A|
> |Synthia to Cityscapes (ours, 16 -> 19)|61.6|72.7|68.8|77.5|23.2|47.0|

---

> ### Author Response · Authors · 2024-11-22
>
> ***Q6: The framework seems to rely purely on SAM for pixel grouping, with UDA and CLIP then reclassifying each mask region. Is this my understanding accurate?***
>
> Thank you for bringing up this question. We would like to clarify that the pixel grouping knowledge in our framework does not rely solely on SAM. The initial pixel grouping is derived from the in-domain predictions of the UDA model. Building on this, SAM is utilized to identify finer-grained pixel groups.
>
> To integrate in-domain knowledge with prior knowledge, we fuse predictions from the UDA model and CLIP to reclassify each region. To enhance the use of prior knowledge, we propose multi-scale visual feature extraction for improved scene understanding at different scales. Additionally, context-aware text feature extraction is introduced to fully capture the semantics of class labels. When the confidence for the reassigned new class is low (below the 0.5 threshold), we retain the pixel grouping from the UDA model's original predictions. Thus, our framework does not rely solely on SAM for pixel grouping with a reclassification based on CLIP. Instead, it combines the strengths of both UDA and SAM predictions to produce a more robust and accurate final pixel grouping. Furthermore, our approach introduces an effective method to fuse knowledge from different sources, rather than merely combining them.
>
> We hope this clarification addresses your concerns and are happy to provide further explanations if needed.
>
> [1] Hoyer, Lukas, et al. MIC: Masked image consistency for context-enhanced domain adaptation. CVPR 2023.
>
> [2] Zhang, Chaoning, et al. Faster segment anything: Towards lightweight sam for mobile applications. arXiv preprint arXiv:2306.14289 (2023).
>
> [3] Wang, Feng, Jieru Mei, and Alan Yuille. Sclip: Rethinking self-attention for dense vision-language inference. ECCV 2025.
>
> [4] Touvron, Hugo, et al. Llama: Open and efficient foundation language models. arXiv preprint arXiv:2302.13971 (2023).

---

> > ### Comment · Reviewer_ddgh · 2024-11-27
> >
> > I carefully reviewed the reviews provided by other reviewers and the authors' rebuttals. The rebuttals effectively addressed most of my initial concerns, leading me to raise my initial score accordingly.

---

> > > ### Author Response · Authors · 2024-11-27
> > >
> > > Thank you for carefully reviewing our work and raising the score! We would like to provide an update with new results on the proposed baseline [B].
> > >
> > > For work [B]: Open-vocabulary SAM (OVSAM) in the context of open-vocabulary segmentation, we thoroughly examined the method and its official codebase. Upon review, we found that OVSAM is not directly applicable to our setting due to its reliance on visual prompts as mandatory inputs, as confirmed in issue #24 of the official repository. Furthermore, OVSAM does not support text prompts as inputs, as noted in issues #22 and #33. In their experiments, OVSAM utilizes proposals derived from ground truth, which is incompatible with our scenario where no labels are available in the target domain. This limitation necessitates the use of an additional proposal framework or ground truth for inference.
> > >
> > > To evaluate the performance of OVSAM under our constraints, following the approach used in Grounded-SAM, we employ Grounding DINO to generate bounding boxes and class proposals. These bounding boxes are then fed into OVSAM to produce segmentation results. The performance of this pipeline is presented in the table below. As shown in the results, our method continues to outperform OVSAM in both known and unknown classes.
> > >
> > > Given OVSAM’s claim of open-vocabulary capabilities, we also conducted experiments where Grounding DINO was used solely for generating bounding box proposals, with OVSAM performing segmentation and classification for each segmented mask. This setup is labeled in the table as _G-DINO + OVSAM (OVSAM for cls)_. Our findings indicate that OVSAM achieves better performance on fine-tuned datasets such as LVIS and COCO while struggling to deliver strong results for open-vocabulary segmentation tasks on the Mapillary dataset.
> > >
> > > We recognize the importance of incorporating additional baselines to enable a more comprehensive evaluation, which will also benefit future research in this field. We are actively exploring other relevant methods and are committed to including further baseline comparisons as we identify suitable works with accessible codebases. Additionally, we welcome any suggestions for correlated works and are more than willing to conduct further studies based on them.
> > >
> > > Once again, we thank you for your invaluable suggestions and feedback, which have greatly contributed to improving the quality of our work!
> > >
> > >
> > > |method|all classes| |known classes| |unknown classes||
> > > |---|---|---|---|---|---|---|
> > > ||mACC|mIoU|mACC|mIoU|mACC|mIoU|
> > > |G-DINO + ovsam (G-DINO for cls)|32.1|26.7|44.4|40.5|23.9|17.5|
> > > |G-DINO + ovsam (ovsam for cls)|15.6|7.4|18.7|9.5|13.4|3.9|
> > > |Ours|53.0| 36.7| 72.6 |62.4 |39.9 |19.6 |
> > > >Note: performance on Mapillarary Vistas dataset. G-DINO denotes Grounding DINO. OVSAM denotes Open-vocabulary SAM. cls denotes classification.

---

### Official Review · Reviewer_Ggqb · 2024-11-04

**Soundness:** 3
**Presentation:** 2
**Contribution:** 4
**Rating:** 6
**Confidence:** 4

**Summary:**

This paper propose two-stage framework called DynAlign for unsupervised taxonomy-adaptive cross-domain semantic segmentation. DynAlign leverages prior semantic knowledge to align source categories with target categories that can be novel, more fine-grained, or named differently. Foundation models are then employed for precise segmentation and category reassignment. Knowledge fusion approach is proposed to dynamically adapt to varying scene contexts.

**Strengths:**

1. The problem that the paper focuses on, taxonomy-adaptive domain adaptation without requiring additional annotations from the target domain, is innovative and interesting.
2. The approach based on the foundation model is explainable, reasonable, and innovative.

**Weaknesses:**

1. The explanation of the cpation in Figure 2 is not clear. Please explain the functions of the two CLIPs in the figure. And the caption introduces "CLIP fusing the visual knowledge from SAM with the semantic knowledge from LLM to reassign accurate labels. The predictions can be used as pseudo-labels to further fine-tune the UDA model". Since the predictions can be used as pseudo-labels to further fine-tune the UDA model, should the second CLIP have an output arrow?  The figure 2 cannot show the process well, which causes some misunderstandings.
2. The paper proposes a two-stage approach. however, three steps are introduced in Figure 2. This makes it difficult to understand the content of figure 2. Please redraw the figure 2 to make the process more intuitive and easy to understand.
3. The proposed method includes many networks and is very complex. Please compare the model's training memory usage, training time, and model parameter.
4. As a new benchmark, will the proposed method publish the complete training and testing code for subsequent follow-up research?
5. In 4.2 SEMANTIC TAXONOMY MAPPING, how is the context description set generated? In formula 2, on which dimension is the average operation performed?
6. The symbol usage is inconsistent. In 4.4 KNOWLEDGE FUSION, the multi-scale visual feature is F_V, while the multi-scale visual feature in Figure 3 is F_g. Please check the symbols and charts in the full text to avoid ambiguity.
7. In Figure 3, what does lane-marking 0.6 mean?
8. In 5.4 ABLATION STUDIES, should multi-scale vision feature be changed to multi-scale visual feature? Please keep the name consistent in the paper.
9. Please add more experimental details. For example, the proposed method uses the CLIP backbone, but is it the CLIP ViT-B-16 or CLIP ViT-L-14 backbone or something else?

**Questions:**

If the rebuttal process can explain the above problems and confusions, I will increase my score.

**Details Of Ethics Concerns:**

As above

---

> ### Author Response · Authors · 2024-11-22
>
> ***Q1-2: The explanation of figure and method can be clearer***
>
>
> Thank you for your detailed suggestions on improving the clarity of our paper.
>
> Regarding the two CLIPs in our figure, we would like to clarify that the first CLIP represents the vision and text encoder used to extract corresponding vision and text features. The second CLIP represents the process of computing similarity between these features to obtain probabilities. We have provided additional details and add arrows in Figure 3 for better understanding in the rebuttal updated version.
>
> For the two-stage approach, as described in _Lines 202–209_, we referred to extracting domain-specific knowledge and prior knowledge. In Figure 2, we depicted three components: extracting domain-specific knowledge, extracting prior knowledge, and demonstrating how they are fused.
>
> We acknowledge that the notations in the figure, along with the explanations, may have caused ambiguity. To address this, we have revised the description into a three-stage framework and updated the figure in the revised version of the paper. Thank you again for bringing this issue to our attention.
>
> ***Q3: Memory usage, training time, and model parameter.***
>
> Thank you for raising this question. Our method comprises three main components: the UDA semantic segmentation model (HRDA), SAM, and CLIP. For semantic segmentation on high-resolution images, the memory usage primarily depends on the semantic segmentation model (HRDA), while the inference time is influenced by the reassignment of novel class labels. We provide detailed information on memory usage, inference time, and model parameters in the table below.
>
> The total model parameter count of our method includes the UDA model parameters, plus 308M for SAM (ViT-L) and 351M for CLIP (ConvNeXt-Large). The reported total memory usage corresponds to the memory allocated during the complete inference process for a single image. We anticipate that further memory optimizations could be achieved by refining the code base, particularly by cleaning up intermediate tensor caches.
>
> Overall, our framework requires reasonable computational resources. Moreover, the UDA framework can be flexibly substituted to accommodate memory limitations. As shown in the table, replacing HRDA with DAFormer significantly reduces memory consumption. For efficient inference, a segmentation model can also be trained using pseudo-labels generated by our framework, as described in *Section 5.3*. For example, a trained Mask2Former model using our inference pseudo labels, indicated as _Ours (pseudo-label, Mask2Former)_, achieves inference in just **0.3 seconds** per single image, offering a significant efficiency boost while maintaining high performance.
>
> |model|memory usage|inference time|model parameter|
> |-----|------|------|------|
> |HRDA| 23.9 GB | 1.9 s | 86 M |
> |Ours (HRDA)|34.8 GB | 64.1 s | 668 M|
> |DAFormer| 10.5 GB | 0.7 s | 86 M |
> |Ours (DAFomer)|21.7 GB | 64.1 s | 668 M|
> |Ours (pseudo-label, HRDA)| 23.9 GB | 1.9 s | 86 M |
> |Ours (pseudo-label, DAFormer)| 10.5 GB | 0.7 s | 86 M |
> |Ours (pseudo-label, Mask2Former)|3.7 GB | 0.3 s | 215 M|
> >NOTE: For the inference time, the performance on average of 10 runs on a single NVIDIA A100-SXM4-80GB is reported.
>
>
> ***Q4: As a new benchmark, will the proposed method publish the complete training and testing code for subsequent follow-up research?***
>
> Thank you for raising this valuable question and recognizing our contribution in proposing a new benchmark! We will release the full training and inference code publicly after publication. We also anticipate that this will encourage and support more follow-up studies in the community.
>
> ***Q5: How is the context description set generated? In formula 2, on which dimension is the average operation performed?***
>
> Thank you for the detailed question! As described in the appendix *A.5 CONTEXT NAMES*, we generate the context descriptions by prompting GPT-4 with the following instruction: *"Generate new class names within the context of street scene semantic segmentation, using the original class name as the head noun. Use synonyms or subcategories of the original class that make sense within this context, and if the class has multiple meanings, add specific context to avoid ambiguity. Please provide the original class names along with the context names."* Using this approach, we ensure that the generated context descriptions are both meaningful and specific to the domain of street scene segmentation.
>
> Regarding *Formula 2*, for each word token in a context description, we compute its CLIP text feature embedding, which has a dimension of [768]. For all context description embeddings, we concatenate them to form a text embedding of shape \([k, 768]\), where \(k\) denotes the number of words in the context description. We then average along the first dimension (\(k\)) to obtain a single averaged representation of shape \([1, 768]\).

---

> ### Author Response · Authors · 2024-11-22
>
> ***Q6: The symbol usage is inconsistent.***
>
> Thank you for reading carefully and pointing this out! We have updated the paper to ensure consistent usage of the term *multi-scale visual feature*. In Figure 3, \( F_l \) represents the local feature, and \( F_g \) denotes the extracted global feature. The final visual feature \(F_V\) is computed as a weighted sum of \( F_l \) and \( F_g \), as detailed in *Formula 6*. To improve clarity, we have added more detailed descriptions to the paper. We appreciate your feedback in helping us make this improvement and are open to any further adjustments that are needed!
>
> ***Q7: In Figure 3, what does lane-marking 0.6 mean?***
>
> Referring to *Formula (7)*, we compute the CLIP visual-text similarity between the visual feature and the text features of the candidate classes. The softmax value of these similarities is used as the prediction confidence for each class label. Specifically, the value *0.6* in Figure 3 represents the estimated softmax similarity for the class label *"lane-marking"*, which we use as the confidence score for predicting the mask region as belonging to this class.
>
> ***Q8: In 5.4 ABLATION STUDIES, should multi-scale vision feature be changed to multi-scale visual feature? Please keep the name consistent in the paper.***
>
> Thank you for pointing out this inconsistency. We have updated all occurrences to ensure uniform usage of the term multi-scale visual feature throughout the paper, including in Section 5.4 and the corresponding figure captions.
>
> ***Q9: Please add more experimental details. For example, the proposed method uses the CLIP backbone, but is it the CLIP ViT-B-16 or CLIP ViT-L-14 backbone or something else?***
>
> Thank you for raising this question. As shown in the table in *Q2*, we use the SAM ViT-L model and the ConvNeXt-Large CLIP model in our experiments. All these implementation details to reproduce our results are included in the instructions and code, which we will make publicly available after publication.

---

> > ### Comment · Reviewer_Ggqb · 2024-11-25
> >
> > There are no major revisions in the current version of the paper, which seems to not fully reflect the comments in the paper. Although some issues have been explained, the presentation of the paper has not been significantly improved. Especially the description of necessary experiments and the revision of figures. Therefore, I keep my score.

---

> > > ### Author Response · Authors · 2024-11-26
> > >
> > > Thank you for your reply! We sincerely apologize for the delay in modifying our paper, as we misunderstood certain aspects of the rebuttal format. To address this, we have uploaded a further revised version of the paper, incorporating additional modifications:
> > >
> > > 1. **Figure 2 Update**: We have updated **Figure. 2** by replacing the general notation "CLIP" with "CLIP encoder" and "CLIP similarity" to better explain its functions. Additionally, we added a red arrow to indicate that the pseudo-label is generated by CLIP and provided further explanations in the figure caption.
> > >
> > > 2. **Three-Stage Method Reformulation**: Thank you again for your suggestion on improving clarity. We have reformulated the method into a three-stage process consistent with Figure. 2: domain knowledge for stage 1, prior knowledge for stage 2, and knowledge fusion for stage 3. These changes can be found in _Lines 202–204, Lines 210–212_ and in the figure caption.
> > >
> > > 3. **Memory Usage and Efficiency Experiment**: The experiment on memory usage and efficiency has been highlighted in ***Section 5.3***, with detailed information included in the appendix ***A.2 COMPUTATIONAL EFFICIENCY*** (_Lines 854–882_).
> > >
> > > 4. **Implementation Details and Code Commitment**: We have added more implementation details in _Lines 401-405_ and included a commitment to making the code publicly available in the **abstract**.
> > >
> > > 5. **Context Names and Text Embedding Explanation**: We highlighted in _Line 284 and Lines 402-403_ that detailed context names are provided in the appendix ***A.8 CONTEXT NAMES*** to make them easier to locate. Additionally, we added an explanation on computing the text embedding average in _Lines 286-289_ for better understanding.
> > >
> > > 6. **Clarification of Visual Features and Notations**: To address potential misunderstandings regarding the different representations of visual features, we added more explanations in _Lines 308-309 and Lines 322-323_. We also included further clarification of important notations in the caption of **Figure. 3** for better readability.
> > >
> > > 7. **Figure. 3 Notation and Explanation**: We updated the notation in Figure. 3 to "CLIP similarity" and added further explanation in the figure caption. Additional clarification has also been added in _Lines 368-370_ to emphasize this detail.
> > >
> > > 8. **Inconsistent Notation Fix**: We addressed the inconsistent notation in _Lines 483-485_ to ensure uniformity throughout the paper.
> > >
> > > 9. **Additional Implementation Details**: We provided more implementation details in _Lines 401-405_. If there are any other crucial details that would be helpful, please let us know, and we will gladly include them.
> > >
> > > All these modifications, along with the detailed revisions you proposed, have been highlighted in blue for easy reference. We deeply appreciate the time and effort you have taken to carefully review our paper and offer such detailed and insightful feedback.
> > >
> > > Please let us know if you feel further changes are necessary to improve the presentation and clarity of the revised version. We look forward to your response and thank you again for helping us enhance our work!

---

> > > > ### Comment · Reviewer_Ggqb · 2024-11-27
> > > >
> > > > I appreciate the author's response. I have seen a lot of improvements in the latest revision of the paper. Note that there is a small problem: A "?" reference in line 404. In addition, my concerns have been addressed, so I will increase my score to 6.

---

> > > > > ### Author Response · Authors · 2024-11-27
> > > > >
> > > > > Thank you for carefully reviewing our revision and raising the score! We also appreciate you pointing out the issue, which we have addressed and corrected in the newly updated version. Thank you again for your support in helping us improve this research.

---

### Official Review · Reviewer_5x3w · 2024-11-04

**Soundness:** 3
**Presentation:** 3
**Contribution:** 2
**Rating:** 8
**Confidence:** 4

**Summary:**

The paper formulates the problem of unsupervised taxonomy adaptive cross-domain semantic segmentation. To bridge both image-level and label-level domain gaps without supervision, the paper proposes a two-stage approach DynAlign that integrates both domain-specific knowledge and rich open-world prior knowledge from foundation models. Experiments on two street scene semantic segmentation benchmarks validates the effectiveness of the proposed method.

**Strengths:**

1. The paper writing is good with well formula representation.
2. The unsupervised taxonomy adaptive cross-domain semantic segmentation task is interesting and meaningful.

**Weaknesses:**

1. In Lines 80-81, the reason why this problem is significant but not be studied needs explain.
2. Experimental results of existing taxonomy-adaptive domain adaptation and UDA methods also need to be presented in the Table 1 for comparison.  More parametric experiments about the confidence in the Lines 372 - 373 need to be conducted and discussed.

**Questions:**

1. In Lines 80-81, why this problem is significant but not be studied?
2. What are experimental results of existing taxonomy-adaptive domain adaptation and UDA  methods on two street scene semantic segmentation benchmarks.  How choose the confidence threshold value in the Lines 372 - 373.

---

> ### Author Response · Authors · 2024-11-22
>
> ***Q1: In Lines 80-81, why this problem is significant but not be studied?***
>
> Thank you for raising this interesting discussion. We find the **unsupervised taxonomy adaptive domain adaptation problem** meaningful yet underexplored, primarily due to its complexity in two key aspects: 1) **Unsupervised Domain Adaptation (UDA):** While UDA methods address domain shifts, they typically assume consistent label spaces, making them inflexible for handling label-space discrepancies.2) **Taxonomy Discrepancies:** Adapting to mismatched label spaces—arising from differences in granularity, definitions, or new categories—remains challenging. Recent open-vocabulary approaches built on foundation models prioritize open-world knowledge but often overlook the integration of domain-specific knowledge critical for specialized tasks.
>
> To conclude, most existing research and benchmarks have tackled these challenges separately (e.g., standard UDA assumes consistent label spaces, while taxonomy adaptation often overlooks domain shifts). Consequently, the intersection of these two problems remains a relatively new area with limited prior work. Our study highlights this gap and aims to provide a flexible framework to address both challenges simultaneously for more robust solutions to this real-world problem.
>
> ***Q2: What are the experimental results of existing taxonomy-adaptive domain adaptation and UDA methods on two street scene semantic segmentation benchmarks?***
>
> Thank you for raising this question. As explained in Lines 410-411, _Traditional DA methods are limited to fixed label spaces, making them inapplicable to our problem setting_. Therefore, we compare the results for known classes of HRDA under traditional UDA setting, adapted from GTA to Mapillary dataset. The results show that our framework effectively preserves domain knowledge for the known classes while incorporating rich prior knowledge for the unknown classes.
>
> We identified two existing taxonomy adaptation methods that address the taxonomy adaptation problem in a few-shot setting [1, 2]. Specifically, [1] focuses on medical semantic segmentation, making it not directly applicable to our experimental setting. [2] requires representative few-shot supervision for each unknown class to adapt the original model. For a fair comparison, we present results for the Synthia → Cityscapes task following the same setting as in [2]. The results demonstrate that our method outperforms on unknown classes, even without in-domain supervision. Additionally, we will attempt to adapt [2] for scenarios involving a larger number of unknown classes on the Mapillary and IDD datasets. These results will be incorporated into the paper to provide a more comprehensive and thorough comparison.
>
>
> |  methods  | all|  known | unknown  |
> |-----------------|-------|-------|-------|
> | HRDA            | N/A     | 65.8 |   N/A    |
> | Ours    |  36.7  |62.4  |    19.6     |
> > Note: Comparison of mIoU between traditional UDA method (HRDA) and Our DynAlign from Cityscapes (19 classes) to Mapillary (45 classes).
>
> |method|all | unknown  |known|
> |-----|----|----|----|
> |TACS|46.3|22.0|50.9|
> |Ours|61.6|23.2|68.8|
> > Note: Comparison of mIoU between the few-shot taxonomy adaptive DA method (TACS) and Our DynAlign from Synthia (16 classes) to Cityscapes (19 classes).
>
>
> ***Q3: How to choose the confidence threshold value in Lines 372 - 373? More parametric experiments about the confidence in Lines 372 - 373 need to be conducted and discussed.***
>
> Thank you for raising this important question. In our experiments, we set the confidence threshold to 0.5 by default for reassigning the class label. To address your concerns, we have conducted ablation studies with varying confidence thresholds, and the results are provided in the table below. The results show that while the mAcc improves with a lower confidence threshold for assigning new class labels, the mIoU may decrease correspondingly. Overall, our framework demonstrates robustness and is not very sensitive to the confidence threshold within a certain value range.
>
> | Threshold | 0.4         | 0.5         | 0.6         | 0.7         | 0.8         |0.9|
> |-----------|-------------|-------------|-------------|-------------|-------------|--------|
> mIoU|35.9|36.7|37.6|36.5|36.6|36.5|
> |mAcc|54.3|53.0|52.1|50.3|47.3|46.5|
>
> [1] Fan, Jianan, et al. Taxonomy adaptive cross-domain adaptation in medical imaging via optimization trajectory distillation. ICCV 2023.
>
> [2] Gong, Rui, et al. Tacs: Taxonomy adaptive cross-domain semantic segmentation. ECCV 2022.

---

### Author Response · Authors · 2024-11-26

We sincerely thank all the reviewers for their time and effort in reviewing our work and for recognizing the significance and practical meaning of the problem addressed in this work. Based on your feedback, we have made further modifications to the revised version of our paper:

1. **Clarity:** We appreciate the reviewers highlighting issues related to clarity and inconsistencies in the writing and figures. In response, we have elaborated further on the methodology, revised the figures, and addressed all the noted minor issues to improve clarity.

2. **Motivation:** We are grateful for the reviewers’ input on emphasizing the main contributions of this work. To address this, we have strengthened the motivation and clearly articulated our key contributions and strengths compared to existing methods. These improvements are reflected in the introduction, related works, and additional analysis in the experiments.

3. **Implementation Details:** We thank the reviewers for acknowledging our contributions and expressing interest in our work. We have added more implementation details in the Experiments section and confirmed our commitment to making the code publicly available in the abstract.

4. **Additional Experiments:** We greatly appreciate the reviewers’ suggestions for inspiring experiments to validate our method further. In response, we have included:
   - **Results under more DA settings:** To demonstrate the practical significance of our work, we conducted additional experiments under traditional UDA, fine-to-coarse adaptation, and taxonomy-adaptive settings with smaller label space gaps. The results are presented in **Table 1** and Appendix **A.5**.
   - **Ablations on each module of our framework:** We evaluated the performance of substituting different modules, as shown in **Table 4** and _Lines 495–501_, to highlight the flexibility of our framework.
   - **Memory Usage and Efficiency:** We computed the memory usage and efficiency of our framework. These results are highlighted in ***Section 5.3***, with detailed information included in Appendix ***A.2 COMPUTATIONAL EFFICIENCY*** (_Lines 854–882_).
   - More additional experiments are detailed in the appendix, and we have addressed individual reviewers' concerns directly. As noted in our previous replies, we will include results from additional comparison methods in the camera-ready version to provide a more comprehensive benchmark and facilitate future research in this area.

We have also made further modifications and elaborations based on individual reviewer suggestions. For your convenience, all changes have been highlighted in blue in the updated revised version.

Once again, we sincerely thank all reviewers for their constructive feedback and invaluable suggestions. Could we kindly request confirmation on whether we have adequately addressed your concerns? We would be more than happy to engage in further discussions or implement additional refinements to improve the paper further based on your feedback.

Yours sincerely,
Authors of Submission 9137

---

### Meta-Review · Area_Chair_RJ89 · 2024-12-16

**Metareview:**

All the 6 reviewers provide positive ratings after rebuttal, with 4 upgraded scores. Initially, the reviewers had concerns about technical clarity/complexity, experimental results with more baselines and realistic settings, usage of foundation models. In the post-rebuttal discussion period, all the reviewers are satisfactory with the authors' comments and revised paper. After taking a close look at the paper, rebuttal, and discussions, the AC agrees with reviewers' feedback of the proposed method being sound and effective to solve a challenging task. Therefore, the AC recommends the acceptance rating.

**Additional Comments On Reviewer Discussion:**

Initially, the reviewers Ggqb, ddgh, cqS3, and 7s4Z focused on concerns about technical clarify, baseline results, realistic settings, and usage of foundation models, respectively. After the rebuttal, the authors have provided more explanations and results while updating the paper actively, in which all the four reviewers are satisfactory with the answers and hence upgraded the scores. The AC considers all the feedback and agrees with reviewers' assessment.

---

### Decision · Program_Chairs · 2025-01-22

Accept (Poster)